# Spatiotemporal Changes in Aridity of Pakistan during 1901-2016

Kamal Ahmed[1, 2], Shamsuddin Shahid[1], Xiaojun Wang[3, 4]*, Nadeem Nawaz[2], and Najeebullah Khan[1]

[1]Faculty of Civil Engineering, Universiti Teknologi Malaysia (UTM), Johor Bahru, 81310, Malaysia
[2]Faculty of Water Resource Management, Lasbela University of Agriculture, Water and Marine Sciences, Balochistan, 90150, Pakistan
[3]State Key Laboratory of Hydrology-Water Resources and Hydraulic Engineering, Nanjing Hydraulic Research Institute, Nanjing, 210029, China
[4] Research Center for Climate Change, Ministry of Water Resources, Nanjing, 210029, China

*Correspondence to*: Xiaojun Wang (xjwang@nhri.cn)

**Abstract.** The changing characteristics of aridity over a larger spatiotemporal scale have gained interest in recent years due to climate change. The long-term (1901-2016) changes in spatiotemporal patterns of annual and seasonal aridity during two major crop growing seasons of Pakistan, Kharif and Rabi are evaluated in this study using gridded precipitation and potential evapotranspiration (PET) data. UNESCO aridity index was used to estimate aridity at each grid point for all the years between 1901 and 2016. The temporal changes in aridity and its associations with precipitation and PET are evaluated by implementing a moving window of 50-years data with 11-year interval. The modified Mann Kendall (MMK) trend test is applied to estimate unidirectional change by eliminating the effect of natural variability of climate, and the Pettitt's test is used to detect year of change in aridity. The result revealed that climate over 60% of Pakistan (mainly in southern parts) is arid. The spatial patterns of aridity trends show a strong influence of the changes in precipitation on aridity trend. The increasing trend in aridity (drier) is noticed in the southwest where precipitation is low during Kharif while a decreasing trend (wetter) in Rabi season in the region which receives high precipitation due to western disturbances. The annual and Kharif aridity is found to decrease (wetter) at a rate of 0.0001 to 0.0002 per year in the northeast while Kharif and Rabi aridity are found to increase (drier) at some locations in the south at a rate of -0.0019 to -0.0001 per year. The spatial patterns of aridity changes show a shift from arid to the semi-arid (wetter) climate in annual and Kharif over a large area while a shift from arid to hyper-arid (drier) region during Rabi in a small area. Most of the significant changes in precipitation and aridity are observed in the years between 1971 and 1980. Overall, aridity is found to increase (drier) in 0.52%, 4.44%, and 0.52% area and decrease (wetter) in 11.75%, 7.57%, and 9.66% area for annual, Rabi and Kharif seasons respectively during 1967-2016 relative to 1901-1950.

## 1 Introduction

More than 20% of the global population is living in arid regions under the threat of severe consequences of climate change, particularly due to increasing hydrological extremes (Alazard et al., 2015). The temporal variability and spatial distribution of precipitation and other hydrological phenomena have significantly changed with the increase in global temperature (Kousari et al., 2014). Changes in precipitation have caused more hydrological extremes such as floods or droughts. The ecosystems of arid and semi-arid climates are sensitive to minor changes in climate (Ahmed et al., 2018). These regions are

also characterised by very complex hydrological systems due to high variability in precipitation which often exhibits extreme behaviours, such as flash floods caused by extreme precipitation and extended droughts due to prolonged dry spell (Buytaert et al., 2012). The droughts are projected to become more frequent and severe in arid regions due to an increase in aridity (Nam et al., 2015) as reported in Iran (Tabari et al., 2012), Serbia (Hrnjak et al., 2014), Turkey (Selek et al., 2018), Iraq (Şarlak and Agha, 2018), India (Ramarao et al., 2018) and China (Liu et al., 2018a) among others. Climate models projected an increase in the range of 11 to 23% by 2100 in global arid and semi-arid climate area which will expand aridification in different parts of the globe (Huang et al., 2016).

Pakistan located in South Asia has a complex terrain with limited water resources. Several attempts have been made to classify the aridity and climate of Pakistan based on different climate variables and methods (Bharuqha and Shanbhag, 1956;Oliver et al., 1978;Shamshad, 1988;Chaudhry and Rasul, 2004;Hussain and Lee, 2009;Zahid and Rasul, 2011;Sarfaraz, 2014;Sarfaraz et al., 2014;Haider and Adnan, 2014). Bharuqha and Shanbhag (1956) classified the climate of a station (Hyderabad) based on the fraction of precipitation to evaporation for the period 1926−1940 and found that Hyderabad has an arid (desert) climate. Oliver et al. (1978) applied clustering approach for climate classification using meteorological data from 53 stations. The results of the study showed that Pakistan has nine climate regimes where most of the area falls under arid climate. Chaudhry and Rasul (2004) used Thornthwaite's precipitation effectiveness index (PEI) for the estimation of annual and seasonal aridity for the period 1961-1990 using temperature data of 50 stations. The results showed that around 75% of the land has arid climate while only a small area in the north-eastern plain has a sub-humid climate. Hussain and Lee (2009) classified the climate using factor and cluster analysis utilising 26 years (1980-2006) rainfall and temperature records of 32 stations. The study concluded that the land of Pakistan could be divided into six regions based on the topology of the country. Haider and Adnan (2014) used several aridity indices (De Martonne Aridity index, Erinc Aridity index, Thornthwaite's PEI, UNESCO Aridity index and Thornthwaite Moisture index) to classify the climate of Pakistan based on records of 54 stations for the period 1961-2009. Their study reported that around 75 to 85% of the land of the country belongs to the arid climate and less than 10% of land in the north belongs to the humid climate. Sarfaraz (2014) used principal component analysis for the sub-regional classification of Pakistan's winter precipitation using 35 station data from 1976 to 2005 and reported six sub-regions of winter precipitation in Pakistan. Sarfaraz et al. (2014) used Köppen classification to classify the climate based on 59 stations data for the period 1981 to 2010 and showed that 75% of the country has arid to semi-arid climate. Recently, Nabeel and Athar (2018) classified the climate based on wet and dry spell using 46 stations data for the period 1976 - 2007. They reported that 66% of the country belongs to the arid climate while only 4% belongs to the humid climate. Even though several studies have been conducted for the classification of climate using aridity indices, there is still no comprehensive study to assess the long-term trends in the aridity of Pakistan in different seasons (annual, Kharif and Rabi). Furthermore, no study has been conducted to determine the impacts of climate change on

aridity, particularly the influence of different climate variables like precipitation, temperature and potential evapotranspiration on aridity in different seasons.

Both increasing and decreasing trend in aridity has been reported in different regions of the world due to climate change. Several studies reported an increase in aridity in global (Dai, 2013;Trenberth et al., 2014) and regional (Ramarao et al., 2018;Jiao et al., 2016) scales. On the other hand, decrease in aridity is also reported in USA (Finkel et al., 2016), China (Yin et al., 2018) and some regions of Iran (Tabari and Talaee, 2013). In recent years, an increase in aridity in some regions of Pakistan has been reported (Haider and Adnan, 2014). However, it was just anticipation based on the assumption that rising temperature has intensified PET and thus an increase in aridity. The magnitude of temperature rises and the changes in regional precipitation pattern determines the changes in the aridity of an area. Therefore, it is required to assess the changes in aridity in regional scale considering the changes in both temperature and precipitation due to global warming.

Several studies suggest rising temperature and changing precipitation in Pakistan due to global warming. Recently, Pakistan experienced several temperature extremes in the form of scorching heatwaves that resulted in significant fatalities. Additionally, the prolonged spell of droughts due to lack of seasonal precipitation has caused enormous economic damages. The annual maximum temperature in the country is increasing at a rate of 0.17- 0.29 °C/decade (Khan et al., 2018), while the precipitation is reported to increase in the north and decrease in the south at a rate of -4 to 4 mm/year in the last fifty years (Ahmed et al., 2017). The variations in temperature and precipitation patterns are also reported in different climatic and cropping seasons (Iqbal et al., 2016). The rising temperature has intensified the evaporation which in turns has caused water losses from major water reservoirs and thus causing water scarcity. In this context, assessing the changing characteristics of precipitation and potential evapotranspiration (PET) over the manifold topography and climate of Pakistan is very important. As the characteristics of precipitation, PET and aridity changes with season and time, it is also imperative to evaluate their changing patterns for different periods and seasons.

The main objective of the present study is to evaluate the changing characteristics of aridity based on precipitation and PET in annual and two distinct cropping seasons (Rabi and Kharif) of Pakistan. Several aridity indices are available for the classification of aridity such as de Martonne aridity index (de Martonne, 1926), Thornthwaite aridity index (Thornthwaite, 1931), Erniç aridity index (Erinç, 1965), UNESCO aridity index (UNESCO, 1979). Among all the aridity indices, the UNESCO aridity index which considers the effect of precipitation and PET for the classification of climate is most widely used (Zarch et al., 2017). The long-term (1901-2016) gauge-based gridded precipitation and PET datasets are analysed by implementing a moving window of 50-years data with 11-year interval. The modified Mann-Kendall trend (MMK) is used to evaluate the significance of changes estimated using Sen's slope estimator and the Pettitt's test is used to identify the year of change in aridity and climate. It is expected that the use of MMK test would provide the changes in aridity due to global warming by eliminating the effect of natural variability of climate which infested as a long-term autocorrelation in time

series. The procedures presented in this study can be used for the assessment of the changing characteristics of aridity and the identification of the factors that drives the changes which can help to understand the possible shift in the climatology of an area owing to climate change. The findings of the study can be helpful for Pakistan in planning adaptation measures and adjust cropping patterns to ensure sustainability in agriculture.

## 2 Study Area and Datasets

### 2.1 Description of the Study Area

Pakistan located in South Asia shares borders with India in the east, China in the north, Iran and Afghanistan in the west, and the long coastline with the Arabian Sea in the south (Fig.1). Around 80% of the land is characterised by arid to the semi-arid climate where precipitation is less and temperature is high (Khatoon and Ali, 2004). The topography of the country varies

widely from plain lands in the south to high mountainous ranges in the north. The large variations in topography from 0 to 8552 m above mean sea level causes a large variation in the climate in the country.

Rabi and Kharif are the two major cropping seasons of Pakistan (Chaudhry and Rasul, 2004). The Rabi season commences in November and finishes in May while the Kharif season starts in April and finishes in October (Nabeel and Athar, 2018). Besides cropping seasons, there are two major rainy seasons, i.e. winter and monsoon which coincide with the Rabi and

Kharif season. Winter precipitation begins in December and lasts till March is important for Rabi crops while Monsoon precipitation begins in June and lasts till September is important for Kharif crops. Winter precipitation occurs due to the moist wind from the Mediterranean Sea in the west and north of Pakistan (Hussain and Lee, 2014). On the other hand, monsoon precipitation occurs due to the moist wind from the Bay of Bengal which contributes 60% of total precipitation of the country (Sheikh, 2001). The agro-economy and the livelihood of farmers constituting 43% of the total population of

Pakistan depend on winter and monsoon precipitation (Ahmed et al., 2018a).

The precipitation in both seasons varies widely in time and space (Ullah et al., 2018b). The mean annual precipitation in Rabi is 119 mm/year, and Kharif is 191 mm/year. The precipitation varies from 10 to 700 mm from the southwest to the north during Rabi and 11 to 900 mm from the southwest to the north near the foothills of Himalaya during Kharif. Most of the country receives precipitation less than < 100 mm in Rabi and < 190mm in Kharif (Ahmed et al., 2018a).

The Rabi and Kharif have contrasting temperature due to their coincidence with winter and summer respectively (Ullah et al., 2018a). The annual average temperature is 14 °C during Rabi and 26 °C in Kharif (Khan et al., 2018). The average temperature in around 10% of the country (the northwest and far north) goes below zero during Rabi while it goes above 30 °C in 45% of the area during Kharif. The overall temperature varies from -12 to 23 °C in Rabi and 1.9 to 33 °C in Kharif (Ahmed et al., 2018a).

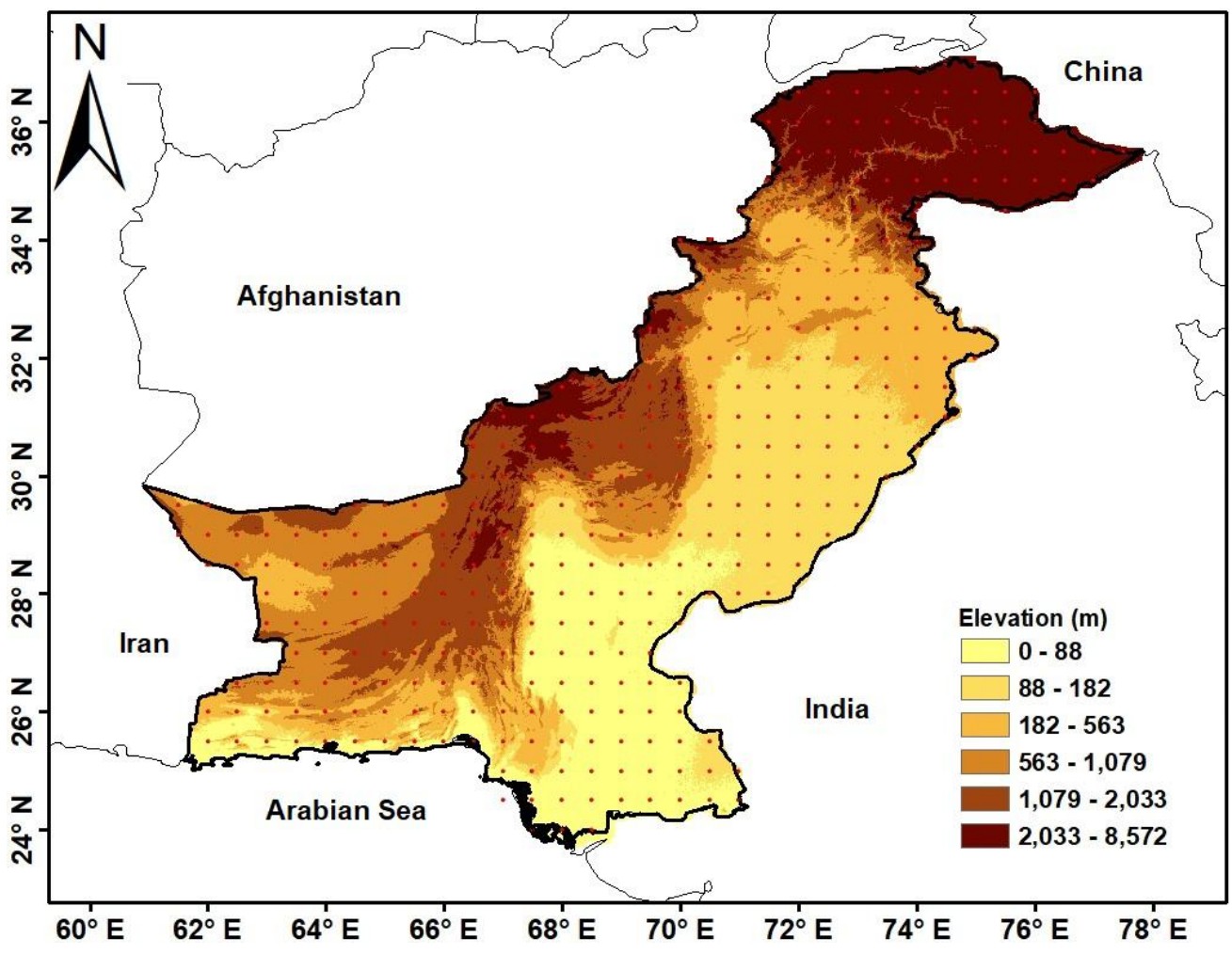

**Figure 1.** Geographic location and topography of Pakistan. The locations of precipitation and potential evapotranspiration data extraction are presented using dots.

## 2.2 Datasets

The gauge-based gridded climate data are widely used as a proxy of observed precipitation and PET data around the world (Shiru et al., 2018). Over the last two decades, several gauge-based gridded datasets have been developed and applied for

different purposes (Li et al., 2014). Among them, the datasets of Global Precipitation Climatology Centre (GPCC) (Schneider et al., 2013) (dwd.de/EN/ourservices/gpcc/gpcc.html) and Climatic Research Unit (CRU) (Harris et al., 2014) of the East Anglia University (crudata.uea.ac.uk) are the most popular due to their spatial and temporal continuity (Kishore et al., 2015). Thus, GPCC precipitation and CRU PET data are used in the present study. The GPCC and CRU data have

several advantages that made the products superior to others. Both the datasets are developed by considering relatively a large number of in-situ data (Schneider et al., 2014;Harris et al., 2014). Besides, these datasets are available at high spatial resolution (0.5°× 0.5°) and longer period (1901 to 2016) that helps better understanding of the changes in climate (Spinoni et al., 2014). Furthermore, an extensive quality control procedure was followed during the development of GPCC and CRU data which has made them more reliable compared to other products (Sun et al., 2014). Additionally, robust interpolation

technique was used in the development of GPCC (Spheremap spatial interpolation) and CRU (Thin plate smoothing splines interpolation) (Becker et al., 2013;New et al., 2002). Several studies revealed better agreement of GPCC and CRU data with station records of Pakistan (Adnan and Ullah, 2015;Asmat et al., 2017). The precipitation and PET data are extracted from 350 grid points for the period 1901-2016 to cover entire Pakistan.

**3 Methodology**

The procedure used for the assessment of the changes in the characteristics of aridity in Pakistan is outlined below:

1) The aridity is estimated as the ratio of precipitation to PET at each GPCC/CRU grid point for all the years during 1901 – 2016. The aridity values are estimated separately for annual, Rabi and Kharif seasons.

2) Sen's slope estimator is used to estimate the rate of change in precipitation, PET and aridity in annual, Rabi and Kharif seasons for the period 1901 – 2016.

3) The MMK trend test is used to evaluate the significance of the change in precipitation, PET and aridity for all the seasons.

4) The influence of precipitation and PET on aridity is assessed for different 50-year with an interval of 11-year over the period 1901 – 2016.

5) The shift in the aridity from one aridity class to another between two periods, 1901 – 1950 and 1967 – 2016 is

mapped to assess the changes in areal extent of different arid classes.

6) The Pettitt's test is used to detect the change points in aridity, precipitation and PET in Pakistan.

### 3.1 Aridity Index

Aridity index (AI) is often used to quantify the long-term climatic conditions of an area (Ashraf et al., 2014). Several definitions of aridity can be found in the literature which are derived using different climate variables like precipitation, temperature and PET (Zarch et al., 2017). Among them, the AI definition of UNESCO (1979) as a ratio of precipitation to PET is most widely used (UNEP, 1992). The precipitation and PET data are averaged for a year or a season to estimate AI. The AI categorised climate of an area into five classes, hyper-arid (AI<0.03), arid (0.03≤AI<0.20), semi-arid (0.20≤AI<0.50), sub-humid (0.50≤AI<0.75) and humid (AI≥0.75). In other words, a higher value of AI indicates wetter while a lower value of AI indicates drier condition.

Various methods are available in the literature to estimate PET. Among them, the Thornthwaite (Thornthwaite, 1948) and Penman-Monteith (Monteith, 1965) methods are the most widely used. The Penman-Monteith method is adopted in UNESCO (1979) while the Thornthwaite method is adopted in UNEP (1992) for defining aridity. Thornthwaite method is preferred over the Penman-Monteith method in data scarce regions (Zarch et al., 2015). However, the Penman-Monteith method provides a better estimation compared to other approaches (Tukimat et al., 2012). Thus, CRU PET data estimated using the Penman-Monteith method is used in the present study.

### 3.2 Sen's Slope Estimator

Sen's slope estimator (Sen, 1968) is a non-parametric method which is widely used for robust estimation of change over a period (Yue et al., 2002;Khan et al., 2018). In this method, the rate of change in data between two consecutive times is first estimated for the whole series. The median of all the consecutive changes in data series is then determined to show the rate of change for the whole period. The slope between two data points is calculated as follows:

$$Q = \frac{x_j - x_k}{j - k} \quad \text{For } i = 1,2,3,.....,n \tag{1}$$

Where $Q$ is the slope between two data points $x_j$ and $x_k$ estimated at time $j$ and $k$. The median of all $Q$ estimated for all the two consecutive data points in the time series is the Sen's slope which gives a measure of change for the whole period.

## 3.3 Modified Mann-Kendall (MMK) Test

In the MMK test (Hamed, 2008), the significance in the trend is first computed by applying classical MK test. The MK test statistics (S) for time series with $n$ data points can be calculated as:

$$S = \sum_{i=2}^{n} \sum_{j=1}^{i-1} Sign\left(x_i - x_j\right) \qquad (2)$$

5    Where $x_i$ and $x_j$ are sequential data and $sign(x_i - x_j)$ is calculated as below:

$$sign(x_i - x_j) = \begin{cases} -1 & for \left(x_i - x_j\right) < 0 \\ 0 & for \left(x_i - x_j\right) = 0 \\ 1 & for \left(x_i - x_j\right) > 0 \end{cases} \qquad (3)$$

The standardised test static ( $\mu_1$ ) is then calculated from the variance of $S$ as,

$$\mu_1 = \begin{cases} \dfrac{S-1}{\sqrt{Var(S)}} & if\ S > 0 \\ 0 & if\ S = 0 \\ \dfrac{S+1}{\sqrt{Var(S)}} & if\ S < 0 \end{cases} \qquad (5)$$

The null hypothesis on no trend is rejected at a confidence interval of 95% if $|\mu_1| > 1.96$ .The MMK test is

10   conducted when the null hypothesis of no trend is rejected. For this purpose, the existing trend in time series data is removed, and the data are ranked. The equivalent normal variants of ranked data ($R_i$) are calculated as,

$$Z_i = \phi^{-1}\left(\frac{R_i}{n+1}\right) for\ i = 1{:}n \qquad (6)$$

15   Where $\phi^{-1}$ is the inverse standard normal distribution function. The Hurst coefficient ($H$) is estimated by maximising the log-likelihood function. If $H$ is found significant, the biased variance of $S$ is calculated as,

$$V(S)^{H'} = \sum_{i<j} \cdot \sum_{k<l} \frac{2}{\pi} \sin^{-1}\left(\frac{\rho_{|j-i|} - \rho_{|i-l|} - \rho_{|j-k|} + \rho_{|i-k|}}{\sqrt{(2-2\rho_{|i-j|})(2-2\rho_{|k-l|})}}\right) \tag{7}$$

Where $\rho$ is the auto-correlation function for given $H$. The unbiased estimate $V(S)^H$ is calculated as,

$$\text{5} \quad V(S)^H = V(S)^{H'} \times B \tag{8}$$

Where $B$ is a function of $H$ as below:

$$B = a_0 + a_1 H + a_2 H^2 + a_3 H^3 + a_4 H^4 \tag{9}$$

Where the coefficients $a_0$, $a_1$, $a_2$, $a_3$, and $a_4$ are the functions of the sample size $n$, which can be found in Hamed (2008). The significance of the MMK test is computed by using $V(S)^H$ in place of $V(S)$ in equation (5).

### 3.4 Relationship of Aridity Trends with Precipitation and PET

15   The relationships of precipitation and PET with aridity are assessed using a moving window of 50-year with 11-year interval over the study period, i.e., 1901-1950, 1912-1961, 1923-1972, 1934-1983, 1945-1994, 1956-2005 and 1967-2016. The main purpose of considering a 50-year window is to decipher the changing pattern in the relationship over the study period. The 11-year interval was considered to assess the relationship for the whole period (1901-2016).

### 3.5 Pettitt Test

The point of change in time series is detected using the Pettitt test (Pettitt, 1979). This nonparametric test allows identification of the point at which any significant shift occurred in time series. The test relies on Mann-Whitney statistic $U_{t,N}$ where the two samples $x_1 \ldots x_t$ and $x_{t+1} \ldots x_n$ are tested to confirm whether they are from the same population or not. Mann-Whitney statistic $U_{t,N}$ is calculated as below:

$$U_{t,N} = U_{t-1,N} + \sum_{j=1}^{N} sgn(X_t - X_j) \ for \ t = 2, \ldots, N \tag{10}$$

and

$$
\begin{aligned}
&if \ (X_t - X_j) > 0 && sgn(X_t - X_j) = 1 \\
&if \ (X_t - X_j) = 0 && sgn(X_t - X_j) = 0 \\
&if \ (X_t - X_j) < 0 && sgn(X_t - X_j) = -1
\end{aligned}
\tag{11}
$$

The test statistic ($K_N$) calculate the number of times where the first sample exceeds the second. The change point is detected when the estimated value exceeds the test statistics ($K_N$), which can be calculated as below:

$$K_N = \max_{1 \leq t \leq N} \left| U_{t,N} \right|$$

$$\tag{12}$$

### 4 Result

### 4.1 Spatial patterns of Precipitation and PET

The spatial patterns in annual and seasonal precipitation and PET for 1901 - 2016 are shown in Figure 2. Precipitation values are grouped in seven classes using natural break algorithm available in ArcGIS 10.3 to show the spatial distribution. Figure 2a shows that mean annual precipitation vary from 38 mm in the south to 2390 mm in the north of Pakistan. The annual average precipitation in the north ranges from 1278 to 2390 mm while it ranges between 38 and 158 in the southwest and some areas in the southeast. The precipitation in Rabi and Kharif seasons are presented as a percentage of annual

precipitation in Figure 2b and 2c respectively. Rabi season coincides with the winter precipitation which enters Pakistan from the west; therefore, Rabi precipitation contributes more than 80% of annual precipitation in the southwest while less than 20% in the southeast (Figure 2b). Kharif season coincides with the monsoon that mostly enters from the northeast of Pakistan. Therefore, the eastern part of the country receives more rainfall in Kharif (60 to >80% of annual precipitation) (Figure 2c). Overall, the average annual and seasonal precipitations are high in the north, low in the southeast during Rabi and in the southwest during Kharif.

Figure 2d depicts the spatial distribution of average annual PET. The PET is relatively high in the south and low in the north. The southwestern and southeastern corners showed the highest PET ranging from 2100 to 2529 mm. The Rabi and Kharif PET as a percentage of annual PET are presented in Figure 2e and 2f respectively. Like the annual, the PET in Rabi and Kharif show more or less similar distributions. In Rabi, PET is low (< 30%) in the north and high (> 45%) in the south while the spatial distribution of PET during Kharif is opposite to Rabi. Relatively lower PET in Rabi indicates the influence of winter (low temperature) while higher PET in Kharif is due to its coincidence with summer when the temperature usually is high in most of the country.

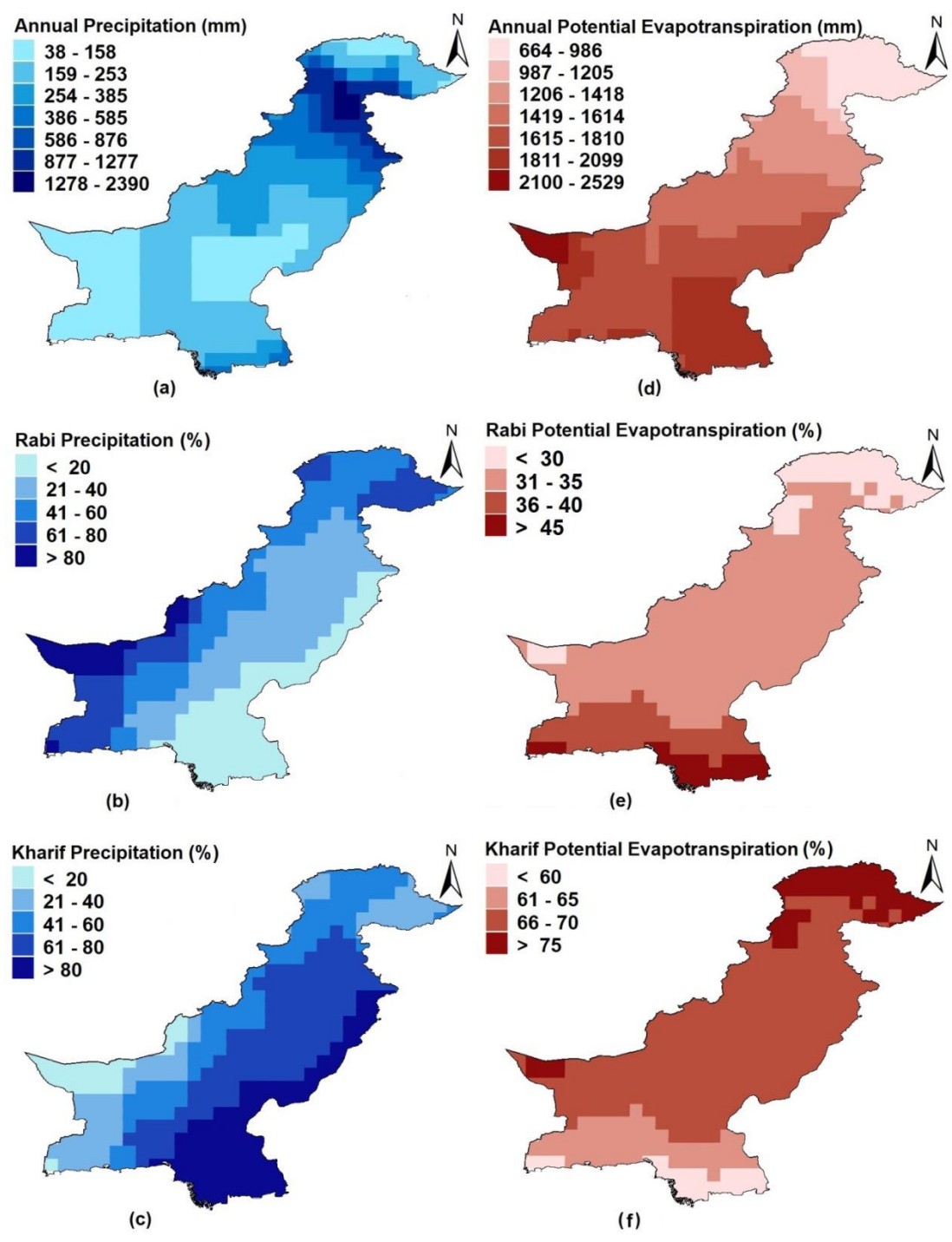

**Figure 2.** Spatial distribution of (a) annual, (b) Rabi and (c) Kharif precipitation; and (d) annual, (e) Rabi and (f) Kharif potential evapotranspiration in Pakistan

## 4.2 Spatial Pattern of Annual and Seasonal Aridity

The AI values are classified as hyper-arid, arid, semi-arid, sub-humid and humid based on UNESCO classification to show the spatial distribution of annual and seasonal aridity in Pakistan (Figure 3). The annual aridity over Pakistan for the period 1901-2016 (Figure 3a) reveals an arid climate in most of the country (61%) followed by semi-arid (21%) and humid (11%). Arid climate covers a larger area in the south and a small area at the top north. The sub-humid and humid climate dominates near the foothills of Himalaya where precipitation is high. On the other hand, the climate in a small area (2%) in the

southwest is found hyper-arid where PET is high, and precipitation is very low.

Figure 3b shows the spatial patterns of aridity during Rabi. Cold winds bring precipitation from the Mediterranean Sea during Rabi season which enters the country from the southwest and therefore, aridity in Rabi is notably less in the southwest. The percentage of the area belongs to semi-arid, sub-humid and humid climate increases during Rabi which indicates a decrease in aridity over a major portion of the country. However, the area belongs to hyper-aridity climate (9%)

increases in the southeast during Rabi. Besides, the aridity in the top north reduces and the humid climate zone near the foothills of Himalaya increases.

Spatial distribution of aridity during Kharif is presented in Figure 3c. Most of the country is characterised by arid climate (59%) followed by semi-arid (20%) and hyper-arid (9%). The area belongs to the hyper-arid climate in the southwest increases during Kharif due to the lack of precipitation in the west during this season. On the other hand, aridity reduces in

the southeast due to monsoon precipitation. The area in the top north and near the foothills of Himalaya which are characterised by semi-arid also reduces which could be owing to an increase in PET.

Overall, figures show that climate in more than 70% of the country is arid to semi-arid. The aridity varies with the season due to the occurrences and dominance of precipitation is different in different seasons. In general, the southern region of the country is characterised by the arid climate and the north is predominantly sub-humid to humid.

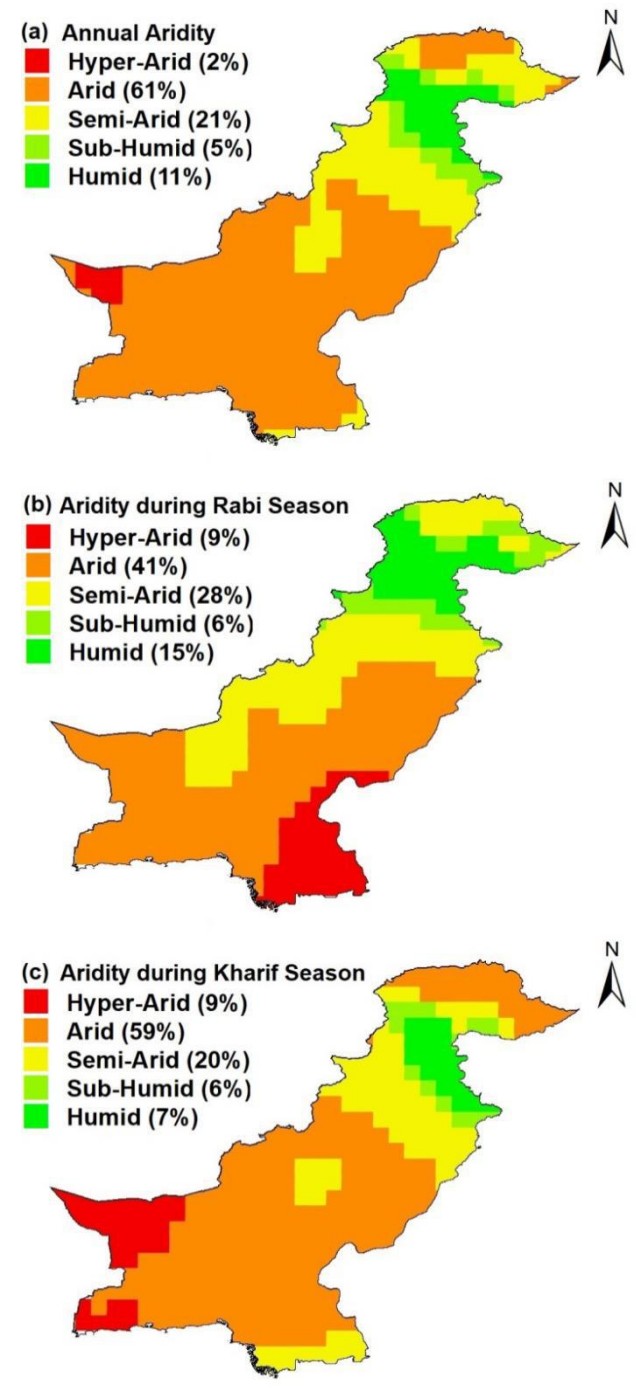

**Figure 3.** Spatial distribution of aridity for (a) annual (b) Rabi and (c) Kharif in Pakistan

## 4.3 Spatial Pattern in the Trends of Precipitation and PET

The sen's slope is used to assess the magnitude of change in precipitation and PET for all the seasons at all the 350 grid
points over Pakistan to prepare the corresponding maps as shown in Figures 4 to 6. The significance increasing/decreasing
trends estimated using MMK test at 95% level of confidence are presented using the plus (+) and minus (-) signs in the
figures. The increase in precipitation indicates a wetter and the decrease a drier condition, while an increase in PET indicates
a drier and decrease a wetter condition. Figure 4a shows that annual precipitation is increasing significantly over a large area
in the northeast and at a few places in the far north, while it is decreasing significantly at a few places in the south and three
locations near the foothills of Himalaya. It is worth to mention that precipitation is decreasing at a few locations near the
foothills of the Himalaya where precipitation is highest in Pakistan (Figure 2a). The spatial distributions of the trends in
annual PET are shown in Figure 4b. The annual PET in Pakistan is increasing (high evaporation rates) in the southeast
corner and decreasing (low evaporation rates) at a few grid points scattered in the center and north-western parts where
precipitation is usually high, and the temperature is low.

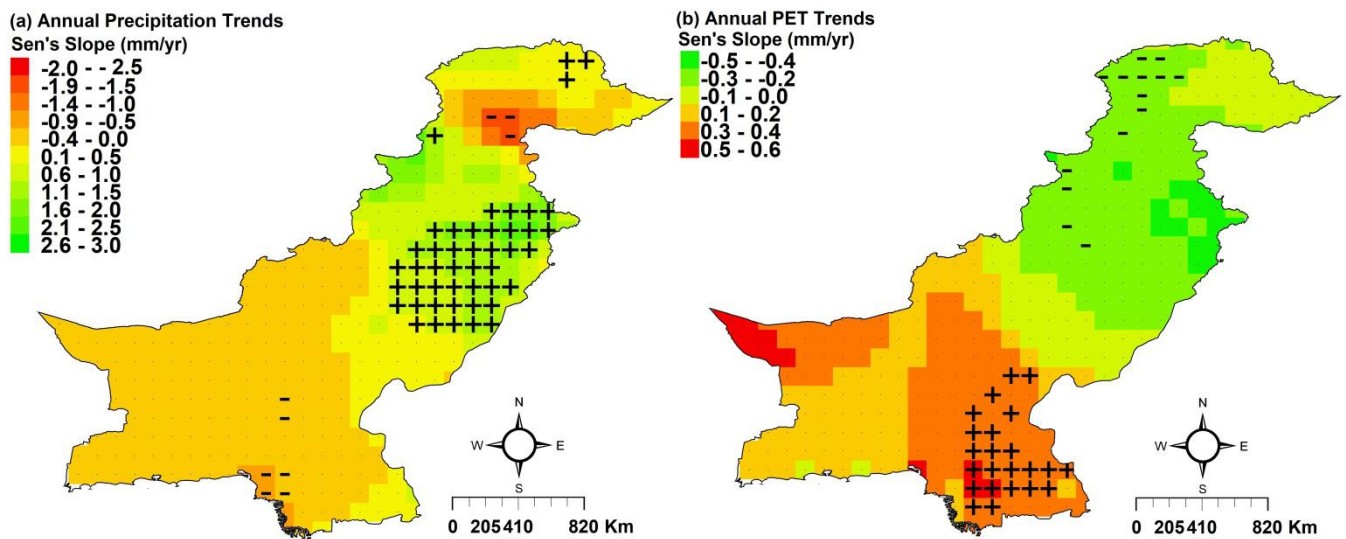

**Figure 4.** Spatial distribution of the trends in annual (a) precipitation and (b) PET in Pakistan estimated using modified
Mann-Kendall (MMK) test. The plus (+) and minus (-) sign indicates increasing and decreasing trend at 95% confidence
level respectively.

Figure 5a shows the spatial patterns in the trend of Rabi precipitation. The precipitation during Rabi is found to increase significantly at a few grid points in the north and two grid points in the east while decreasing significantly at two locations in the south. It can be observed that there is a non-significant decreasing tendency in Rabi precipitation over a large in the south. The PET in Rabi (Figure 5b) is found to increase significantly (high evaporation rates) over a large area in the southeast and the southwest, while it is not found to decrease significantly at any location.

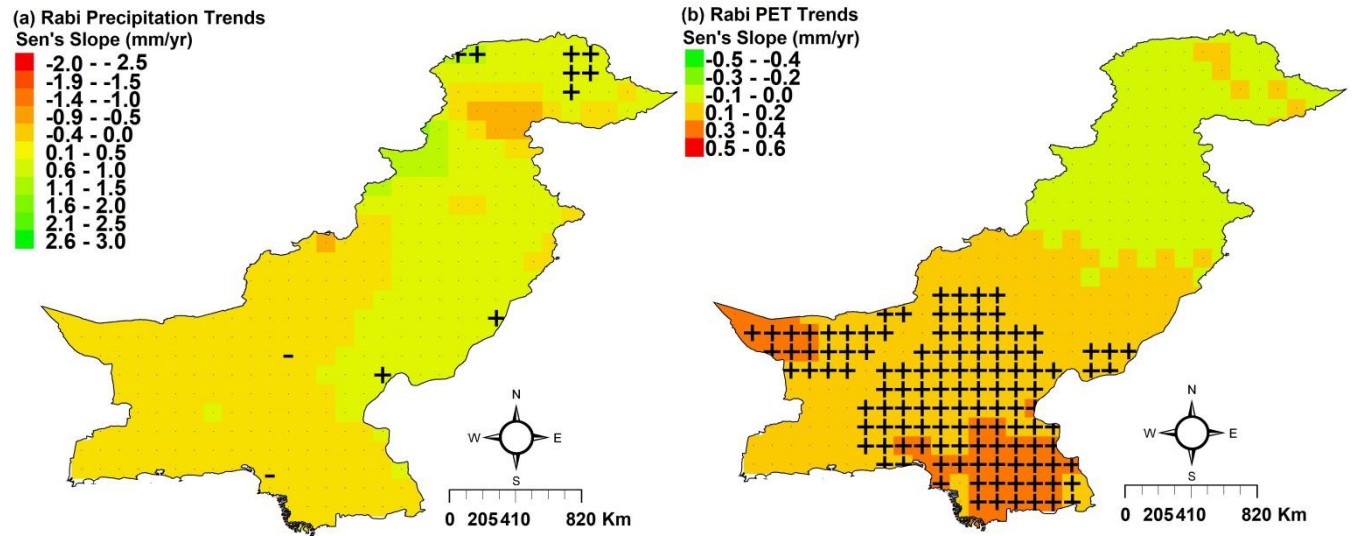

**Figure 5.** Spatial distribution of the trends in Rabi (a) precipitation and (b) PET in Pakistan estimated using modified Mann-Kendall (MMK) test. The plus (+) and minus (-) sign indicates increasing and decreasing trend at 95% confidence level respectively.

The Kharif precipitation (Figure 6a) is found to increase significantly in the northeast and at two grid points in the north. The significant decreasing trend in Kharif precipitation is also observed over a large area in the southwest and at a few grid points near the foothills of Himalaya. Overall, the spatial patterns in annual and Kharif precipitation trends are found very similar. The spatial distribution of PET trends in Kharif is displayed in Figure 6b. The figure shows a significant decrease in PET over a large area in the northeast and decreases at two grid points in the south.

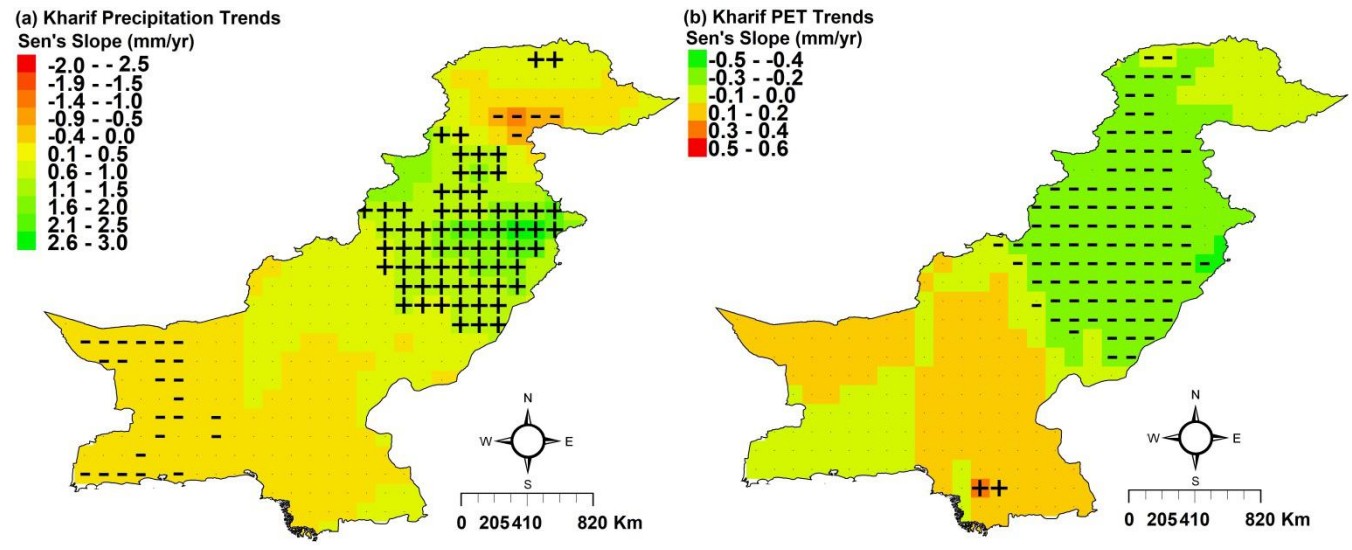

**Figure 6.** Spatial distribution of the trends in Kharif (a) precipitation and (b) PET in Pakistan estimated using modified Mann-Kendall (MMK) test. The plus (+) and minus (-) sign indicates increasing and decreasing trend at 95% confidence level respectively.

## 4.4 Spatial Pattern in the Trends of Annual and Seasonal Aridity

The Sen's Slope method was used to estimate the changes in aridity values calculated using UNESCO method and the MKK test was used to determine the significance of the change at 95% level of confidence. The changes in aridity index are found in the range of -0.0039 to 0.0060 for Pakistan (Figure 7). The values were divided into five classes using natural break algorithm available in ArcGIS 10.3. The plus sign in the figure indicates a significant reduction in aridity (wetter condition) while the minus sign indicates a significant increase in aridity (drier condition). Figure 7a shows that the mean annual aridity has a significant wetter trend over a large area in the northeast and a significant drier trend at a few locations in the south. The aridity trends in Rabi (Figure 7b) shows a significant drier trend in the southwest, at two grid points in the center and in the south. Significant wetter conditions during Rabi are also observed at a few grid points in the north. The aridity trend in Kharif (Figure 7c) is found to follow similar patterns of annual aridity trend. Significant wetter tend is noticed over a major area in the northeast and at a few grid points in the north while drier trend at few locations in the southwest and southern

corner of the country. Overall, the results reveal a wetter trend over a major portion in the northeast and drier trend at few locations in the southwest.

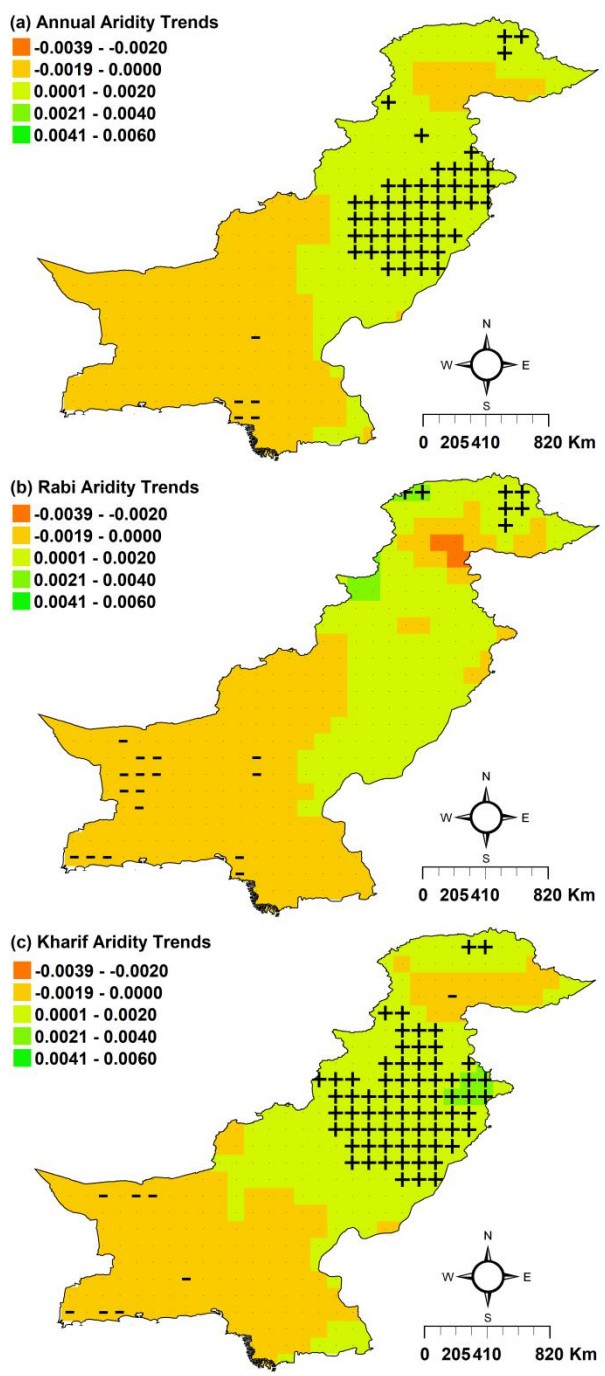

**Figure 7.** Spatial distribution of the trends in (a) annual (b) Rabi and (c) Kharif aridity over Pakistan estimated using modified Mann-Kendall (MMK) test. The plus (+) and minus (-) sign indicates increasing and decreasing trend at 95% confidence level respectively.

**4.5 Time-varying Trends in Areal Extent of Aridity**

A moving window of 50-years with 11-year interval over the period 1901-2016 is used to assess the time-varying trends in aridity, precipitation and PET. The major purpose was to understand the influence of precipitation and PET on aridity in different periods. The obtained results are presented in Figures 8 and 9. The figures show a higher influence of precipitation on aridity compared to temperature. For instance, a reduction in precipitation at 80 grid points caused an increase in aridity at 10 77 grid points in 1923-1972. On the other hand, a decrease in PET at 150 grid points was the reason for a reduction of aridity at 40 grid points during 1934-1983 (Figure 8). Similar results are also noticed for Rabi and Kharif seasons (Figure 9). Therefore, it can be remarked that the changes in precipitation have a higher impact compared to PET in determining the aridity of Pakistan.

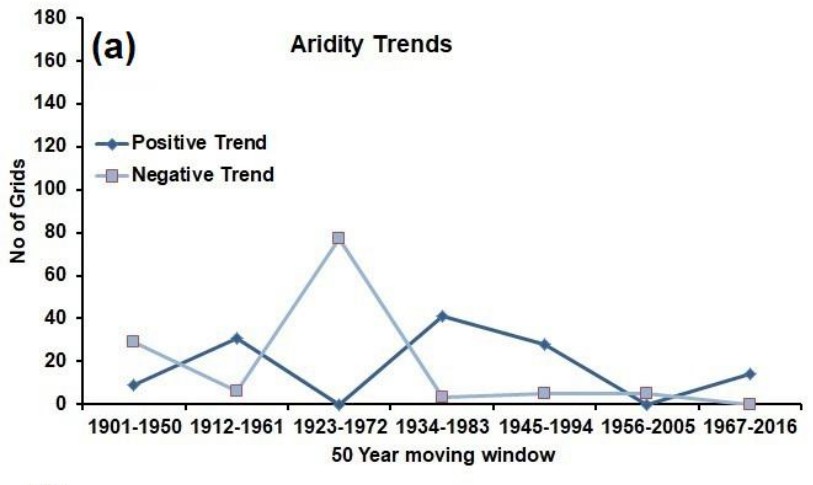

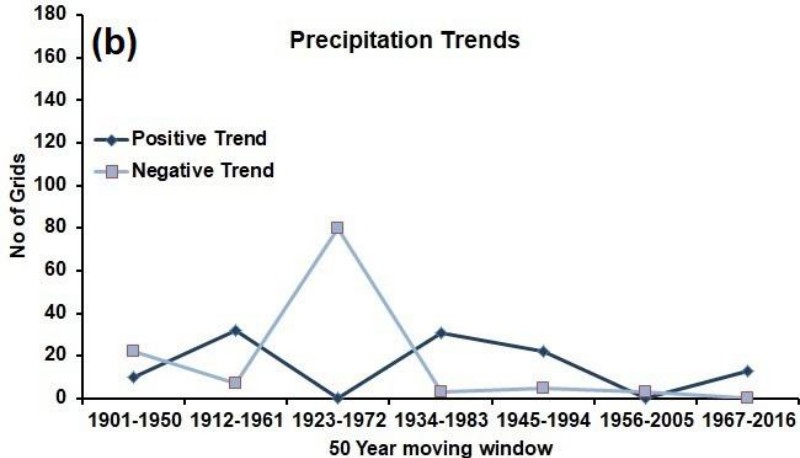

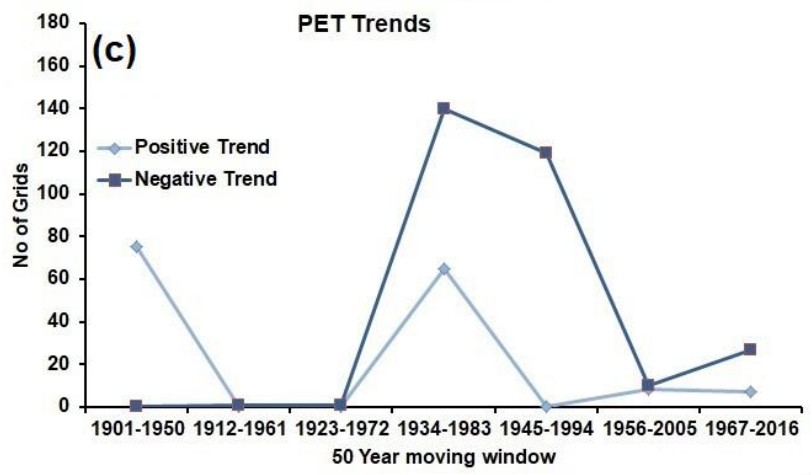

**Figure 8.** Number of grids where annual aridity, precipitation and PET are changed significantly during different periods

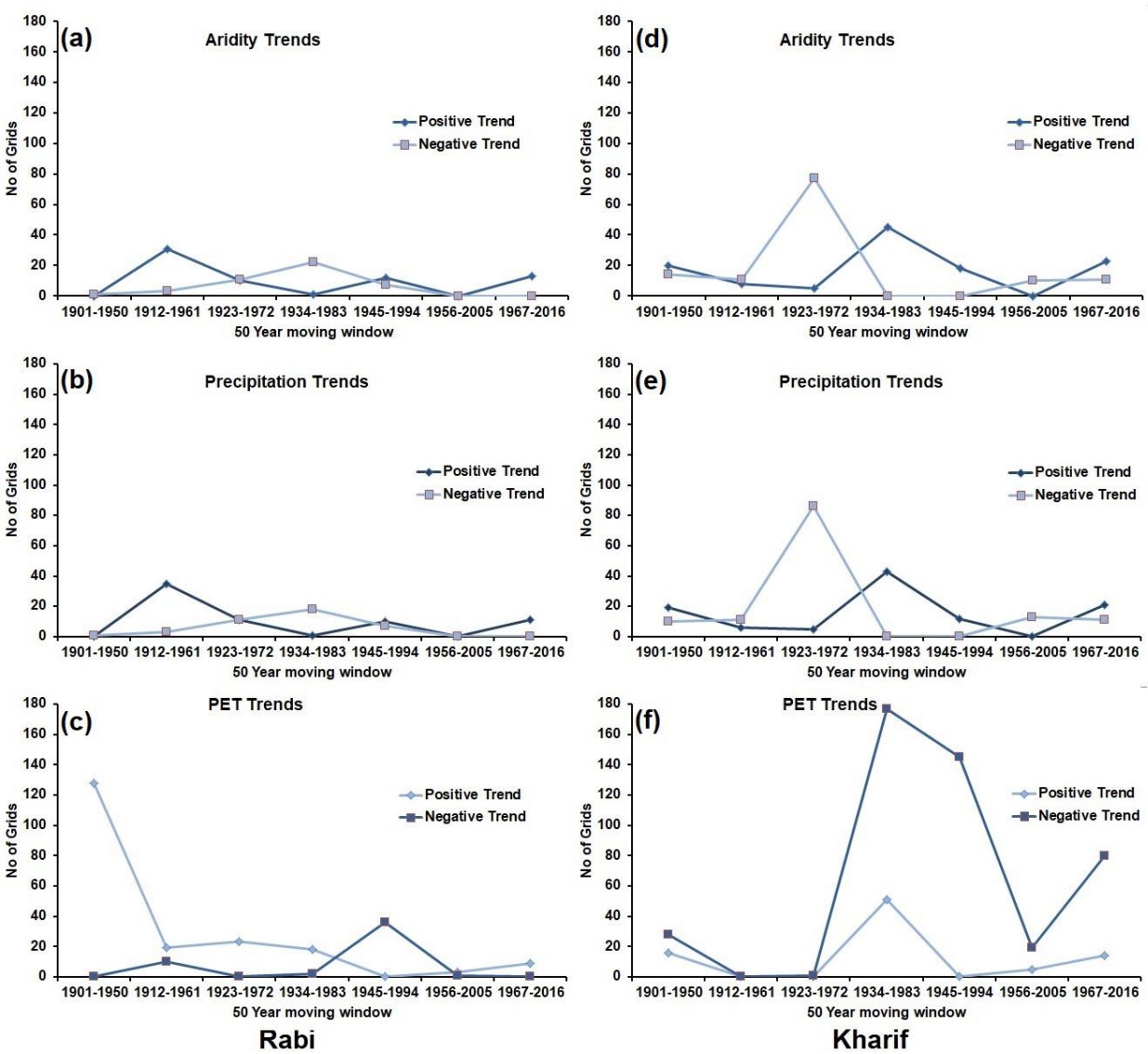

**Figure 9.** Number of grids where annual Rabi and Kharif aridity, precipitation and PET are changed significantly during different periods

## 4.6 The Shift in Aridity

The spatial pattern in the shift of aridity from one class to another is estimated by comparing the aridity maps of the early period (1901-1950) and late period (1967 to 2016). The obtained results are presented in Figure 10. The shifting of aridity from one to another class are illustrated using different colours while the white colour represents no shift in aridity class. The annual climate in a large area is found to shift from arid to semi-arid (Figure 10a). A shift from semi-arid to sub-humid climate is also observed at a few grid points near the foothills of Himalaya. On the other hand, the climate at two grid points in the southwest is found to shift from semi-arid to arid.

Relatively more changes in aridity during Rabi (Figure 10b) compared to annual is observed. A large area in the southeast has changed from arid to hyper-arid. The climate at some grid points in the center and the southwest are also found to change from semi-arid to arid. Besides, the sub-humid climate at a grid point in the north is found to become humid. The climate at several points is also changed from hyper-arid to arid in the southeast and arid to semi-arid at different locations in the north. The spatial pattern of the shift of climate in Kharif (Figure 10c) reveals a change in arid to the semi-arid climate in the central region and hyper-arid to arid at a grid point in the southwest corner while semi-arid to arid at a grid point in the southeast corner.

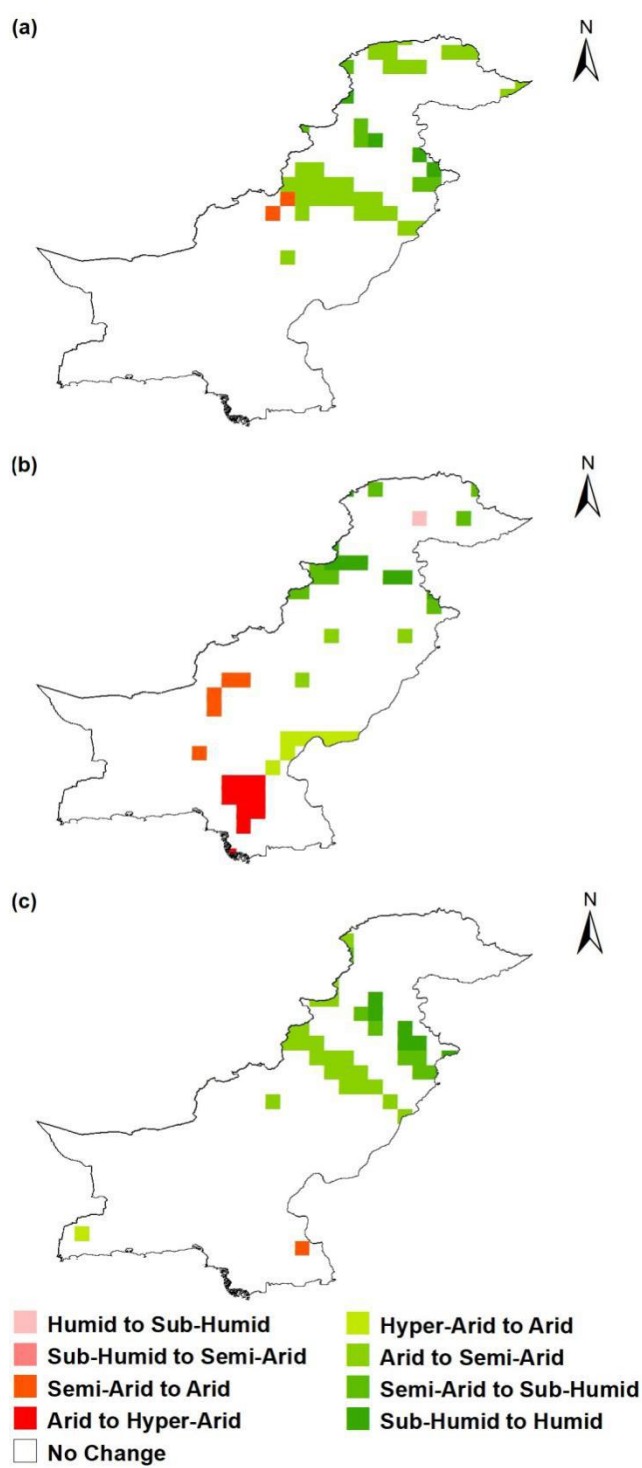

**Figure 10.** Changes in the spatial patterns of aridity between 1901-1950 and 1967-2016

The percentage of changes in different aridity classes are shown in Table 1. No shift in aridity class is observed in more than 85% of the area. There are both positive shift (more arid to less arid class) and negative shift (less arid to more arid class). However, positive shifts are found relatively more compared to negative shifts.

The highest positive shift is found from arid to semi-arid climate (9.14% of the total area for annual and 5.48% for Kharif) while 2.61% area is noticed to shift from semi-arid to sub-humid climate during Rabi season. On the other hand, a negative shift in only 0.52% and 0.27% areas are noticed for annual and Kharif and relatively in a higher area (2.61%) for Rabi.

**Table 1.** Percentage of the area shifted from one aridity class to another between the periods 1901-1950 and 1967 to 2016.

| Class Changes | Annual | Rabi | Kharif |
|---|---|---|---|
| H ➔ SH | 0.00 | 0.26 | 0.00 |
| SH ➔ SA | 0.00 | 0.26 | 0.26 |
| SA ➔ A | 0.52 | 1.31 | 0.27 |
| A ➔ HA | 0.00 | 2.61 | 0.00 |
| NC | 87.73 | 87.99 | 89.82 |
| HA ➔ A | 0.00 | 2.09 | 0.78 |
| A ➔ SA | 9.14 | 0.78 | 5.47 |
| SA ➔ SH | 1.57 | 2.61 | 1.83 |
| SH ➔ H | 1.04 | 2.09 | 1.57 |
| Increase in Aridity (Drier Condition) | 0.52 | 4.44 | 0.52 |
| Decrease in Aridity (Wetter Conditions) | 11.75 | 7.57 | 9.66 |

H: Humid; SH: Sub-Humid; SA: Semi-Arid; A: Arid; HA: Hyper-Arid; NC: No change

## 4.7 Detection of Change Point in Climate

The areal averages of aridity, precipitation and PET of different aridity classes are used to detect the year of their changes using Pettitt's test. The significant changes detected in different years are presented using bold letters in Table 2. Most of the changes in aridity and precipitation are detected between 1971 and 1980 while the change point for PET showed more significant changes compared to aridity and precipitation.

It is important to note that the changes (years) detected for aridity and precipitation are the same for all seasons. For example, the change point of both aridity and precipitation in the hyper-arid region is 1983. The results again suggest that the influence of precipitation on aridity is higher compared to PET.

**Table 2.** The year of change in aridity, precipitation and PET in different climatic regions of Pakistan. The bold number in the table represents the year of significant change

| Season | Class | Aridity | Precipitation | PET |
|--------|-------|---------|---------------|-----|
| Annual | Hyper-Arid | 1983 | 1983 | **1957** |
|        | Arid | 1974 | 1974 | 1967 |
|        | Semi-Arid | **1974** | **1974** | **1974** |
|        | Sub-Humid | **1971** | **1971** | **1963** |
|        | Humid | 1997 | 1997 | **1974** |
| Rabi   | Hyper-Arid | 1975 | 1975 | **1940** |
|        | Arid | 1961 | 1961 | **1939** |
|        | Semi-Arid | 1978 | 1978 | 1939 |
|        | Sub-Humid | 1980 | 1980 | 1974 |
|        | Humid | 1977 | 1978 | 1974 |
| Kharif | Hyper-Arid | 1948 | 1948 | **1965** |
|        | Arid | 1974 | 1974 | **1948** |
|        | Semi-Arid | **1974** | **1974** | 1954 |
|        | Sub-Humid | 1952 | 1952 | **1963** |
|        | Humid | 1952 | 1952 | **1954** |

## 5 Discussions

The changes in aridity depend on the changes of different climatic variables. The present study found precipitation as the most dominating factor to drive the changes in aridity in Pakistan. Several literatures are available on the changes in aridity in the neighbouring country of Pakistan namely, India, China and Iran. The influence of different climatic variables is examined in those studies to identify the driving factors of aridity changes. Ramarao et al. (2018) assessed the changing pattern of aridity in the semi-arid regions of India during 1951–2005 and reported an increase in semi-aridity in the last decade due to the reduction in precipitation and escalation of PET. Ramachandran et al. (2015) assessed the changing behaviour of aridity in the East Coast of South India using regional climate models and RCP4.5 scenario. They reported aridity would increase with the increase in temperature and lowering of precipitation; however, the rising temperature has more influence on the aridity in the East Coast of South India. Gao et al. (2015) evaluated the relationship of aridity with PET, precipitation, temperature, sunshine duration, wind speed and diurnal temperature range over the Tibetan Plateau and found precipitation as the most dominating factor that contributes to the aridity. Liu et al. (2018b) assessed the individual contribution of different variables including precipitation, temperature, wind speed, sunshine duration, and total solar radiation on aridity of China for the period 1961-2006 and showed that the contribution of different variables on aridity varies from region to region, but precipitation is the most dominating factor in most of the regions. Tabari and Talaee (2013) reported increasing PET and decreasing precipitation are the cause of increasing aridity in Iran. Most recently, Araghi et al. (2018) also identified increasing temperature and decreasing precipitation due to global warming as the major cause of increasing aridity in Iran. These studies indicate different climate variables as the major driver of aridity in the region. The present study reveals changes in precipitation are the major cause of the changes in aridity in Pakistan.

Pakistan receives precipitation from the monsoon originated in the Bay of Bengal and the western disturbances originated from the Mediterranean Sea. Monsoon contributes a large quantity to annual precipitation as compared to winter rainfall (Sheikh et al., 2009). Therefore, the geographical distribution of annual precipitation is found more or less the same with the monsoon. Several studies such as Ahmed et al. (2017) claimed that climate change has altered monsoon precipitation in the form of more precipitation in the north and at a few places in the southeast of Pakistan. A similar pattern in annual and Kharif precipitation trends has been observed in the present study. The aridity has decreased in the area where precipitation has increased. The PET is found to increase significantly in a large area in the southeast, but its impact is not significant for annual aridity. Like monsoon, an increase in winter precipitation in a large area has been reported (Ahmed et al., 2017). The aridity during Rabi season is found to follow the same pattern of Rabi precipitation. However, a mismatch in rainfall and aridity trends is found in the southwest. This is due to a large increase in PET in the region. Khan et al. (2018) reported a rapid rise in temperature in the southwest which has probably increased PET and aridity in the area. This indicates that both the changes in precipitation and PET have impacts on the changes in aridity in Pakistan. However, precipitation has a much higher influence on the aridity of Pakistan compared to PET.

The aridity is found to increase (drier) and decrease (wetter) in different regions and seasons with the changes in precipitation and PET. Overall, 11.75%, 7.57%, and 9.66% areas are found to shift to wetter while 0.52%, 4.44%, and 0.52% areas to drier condition for annual, Rabi and Kharif respectively. It is important to mention that a large area has a wetter trend in recent years particularly in the semi-arid or sub-humid regions which mean more area become wetter in recent years. However, some areas in the arid region are found to become drier. This indicates that few dry regions are becoming drier and a large relatively wet area is becoming wetter. A similar finding has been reported by Liu et al. (2018b) in neighbouring China. Overall, a large area in the northeast of Pakistan has become wetter and a few locations in the south become dried during 1901 - 2016.

Pakistan is mainly an agriculture-based country where a notable portion of the population is associated with the agro-based economy. Haider and Adnan (2014) reported that changes in aridity could have a severe impact on the agricultural sector of Pakistan. They showed that some regions in the northeast of the country are becoming less arid while some of the regions in the south are becoming drier. It is pertinent to mention that southern regions of the country are highly prone to droughts (Ahmed et al., 2018b). Increase in drier conditions can have a severe impact on the agricultural-based economy of the south. Similarly, the agriculture of north-eastern regions can be benefitted by the wetter condition.

The changes in temporal patterns of aridity reveal that the major shift in aridity and rainfall occurred between 1971 and 1980. Global atmospheric moisture amount is found to increase after 1973 (Ross and Elliott, 2001). An increase in precipitation in many regions of the world is observed due to the increase in global moisture content (Trenberth, 1998). The present study suggests that precipitation of Pakistan has also changed during 1971-1980 which may be due to the increase in global atmospheric moisture after 1973. This has caused a shift in precipitation and aridity in Pakistan. Machiwal et al. (2017) reported a significant change in dry season precipitation in the period 1973–1975 in the hot arid region of India. Some'e et al. (2012) assessed the change points in precipitation in the eastern part of Iran bordering Pakistan and reported a shift in annual precipitation at some stations during 1981-1982. The results collaborate with the finding of precipitation and aridity shift in Pakistan.

Many factors influence regional and local changes in precipitation including shift in monsoon circulation due to global climate change (IPCC, 2014), land use changes like the changes in forest cover and irrigated agriculture (Pielke, 2001) and aerosols in the atmosphere due to human activities (Guo et al., 2016). Studies related to anthropogenic activities on precipitation changes in Pakistan and nearby countries are very limited (Basistha et al., 2009). Previous studies suggested that global warming as the cause of the shift in precipitation pattern in the region (Duan et al., 2002; Gautam et al., 2009). The nature of the shift in rainfall regime over a large region which coincides with the increase in global atmospheric moisture suggests that global climate change may be the cause of the shift in precipitation and aridity of Pakistan.

The present study suggests that the relative influence of precipitation and temperature on aridity determines its trends in the context of climate change. Aridity may decrease due to a small increase in precipitation in the regions where the influence of precipitation is higher on aridity. The gridded data used in this may cause uncertainty in the estimation of aridity and its trends. Other gridded data can be used in future to assess the uncertainty in the estimated trends in aridity. Besides, different aridity assessment methods can be used to compare the results.

## 6. Conclusions

The long-term changes (1901-2016) in annual and seasonal aridity in Pakistan and its causes are analysed in this paper. Gauge-based gridded precipitation and PET data are used to show the spatial and temporal patterns of the changes in aridity over the diverse climate of the country. Following conclusions are drawn based on the findings: (1) The precipitation is high in the north and low in the southeast and southwest during both Rabi and Kharif seasons while the PET is low in the north due to the cold climate and high in the south due to high temperature. (2) Most of the country is characterised by arid and semi-arid climate except the northern region near the foothills of Himalaya which is characterised by sub-humid to the humid climate. However, the aridity of the country is found to vary for different seasons due to the spatial pattern of precipitation occurrence in the corresponding season. (3) The annual and Kharif precipitation of Pakistan is increasing in the northeast, while Rabi precipitation is increasing at a few grid points in the north. The decreasing trends in annual and seasonal precipitation are mostly observed in the southern parts of the country. (4) The increases in annual and Rabi PET are noticed in the southeast corner while a decrease in Kharif PET over a large area in the north. (5) The aridity for annual and Kharif showed wetter trends over a large area in the northeast and drier trends at a few points in the south while the aridity in Rabi showed drier trends in the southwest and wetter trend in a small area in the north. (6) Overall, there is a wetting tendency over a large area in the northeast and drying tendency at few locations in the southwest. Therefore, it can be remarked that Pakistan has become wetter from 1901 to 2016. (7) The time-varying trends in aridity reveal that the influence of precipitation is high on the aridity compared with PET. Increase in precipitation in the southeast has reduced the aridity to some extent in the region. Even though the increasing temperature has caused an increase in PET, but its influence is found less on aridity. (8) The changes in spatial patterns of aridity show that the climate in a large area has shifted from arid to semi-arid for annual and Kharif while a small area from arid to hyper-arid in Rabi. (9) The highest shift in arid climatology is observed between arid to semi-arid. About 9.1% area is found to shift from arid to semi-arid climate between the periods 1901-1950 and 1967 to 2016. (10) A significant shift in aridity and precipitation in most of the climatic regions of Pakistan is found during 1971-1980.

*Data availability.*

The model codes and the data are available upon request.

*Author contributions.*

KA, SS, and XW designed the research and wrote the manuscript. NN and NK critically reviewed the paper.

*Competing interests.*

The authors declare that they have no conflict of interest.

*Acknowledgement*

We are grateful to the developers of GPCC, and CRU for providing gridded precipitation and PET datasets.

*Financial support.*

This work is supported by the Young Top-Notch Talent Support Program of National High-level Talents Special Support Plan and Professional Development Research University (PDRU) grant no. Q.J130000.21A2.04E10 of Universiti Teknologi
Malaysia.

*Review statement.* This paper was edited by Nadia Ursino and reviewed by three anonymous referees.

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
