# Peer review of "Spatiotemporal Changes in Aridity of Pakistan during 1901-2016"

_Hydrology and Earth System Sciences, 2018_

## Referee Comment (RC1) · Anonymous Referee #1 · 13 Feb 2019

The present article deals with aridity in Pakistan, certainly an important issue that merits publication in HESS. However, in its present form, this MS has too many drawbacks and a very limited benefit to the readers. This is due to several reasons such as, confusing definitions of the two seasons Rabi and Kharif (p.23, l.21-25). Once Rabi is defined as Nov.-May and later as Dec.-Mar.; Kharif is defined as Apr.-Oct. and then as Jun.-Sep. To which definition one should refer? Non-comparable presented maps and partial information in some charts. Figure 1 - I assume that the six different height categories were selected in order to present equal areas in each category. This caused that range of each category is arbitrary and unusual. However, I could cope with this figure as we don't have to compare it to other maps. It is more severe with the rest of the maps. Figure 2 - The same problem as with the previous. It is impossible to

compare among the precipitation maps (2a-2c) or the PET maps (2d-2f) in the different seasons. Furthermore, I suggest the authors to present the precipitation and the PET in the different seasons (2b-2c and 2e-2f, respectively) as a percentage of the annual totals and not as absolute values. This will be more explicit. In their present form, these maps are completely useless. Section 4.5 and Figure 6 and 7 – The authors present a moving average of 11 years of aridity, precipitation and PET trends. I don't understand why they chose increments of 50 years, why not calculate a moving average of 11 yrs. (or any other duration) for the entire period? Such a calculation would result in a less "fuzzy" behavior of the trends and enable to better locate the drier or wetter periods. Apparently there is no reason to assume that trends change with increments of 50 years. Furthermore, a comparison of Figures 6 and 7 reveals that the trends during Kharif are by far more important in determining the entire year trends. Figure 7 is misleading as the vertical axes of a, b and c (Rabi) are different from those of d, e and f (Kharif) and give the impression of trends of the same order of magnitude in both period, which is not the case. The results presented in Figure 8 and Table 1 contradict the postulated in the introduction regarding increased aridity in Pakistan (p.2) and the results cited from Haider and Adman (2014). Overall there is no tendency over the majority of the territory (88%) and in those regions with a tendency it is mainly towards a reduction in aridity. Therefore, large parts of the introduction are irrelevant. Concluding that aridity depends mainly on precipitation (p.23, l-14-15) is very trivial. The authors should consider editing the text by a native English speaker.

---

## Referee Comment (RC2) · Anonymous Referee #2 · 13 Mar 2019

This paper analyzes spatial patterns of aridity across Pakistan and attempts to attribute spatial and temporal changes in these patterns to changes in precipitation and potential evapotranspiration (PET). The paper shows that Pakistan is becoming less arid over a large region (the Northeast), most likely due to increases in precipitation, which seems to contrast with the findings of previous studies.

Overall, I found the variety of statistical tests performed an interesting way to attribute the change in aridity to changes in precipitation with some degree of confidence. However, I was surprised that the paper failed to emphasize the clear geographic correspondence between changes in aridity and precipitation trends shown in Figures 4-5.

My biggest criticism of the paper is the lack of clarity in describing the physical meaning of the results. An increasing aridity index (AI) that indicates decreasing aridity is

confusing enough. Add to that the desire to communicate changing trends in AI, and one can see how a reader becomes quickly confused. The paper would be greatly improved if it simply stated from time to time whether the results indicate that Pakistan is become more or less arid. There are also a number of places (noted below) where figure captions should contain more detail.

My second biggest concern is that the paper fails to discuss its main finding, that Pakistan has become less arid, in the context of previous studies, which largely indicate that Pakistan has recently dried. The fact that this result seems to contrast with previous investigations ought to be discussed.

More specific comments follow:
* * *
INTRODUCTION: The second paragraph of the Introduction lists many studies that have evaluated Pakistan's climate. A brief synopsis of their findings (not just their methodology) seems warranted.

Furthermore, it seems a bit contradictory to end the second paragraph by saying that "no attempt has been made...to assess the changing characteristics of arid climatology..." and then begin the third by saying "In recent years, an increase in aridity is reported..." Clearly, someone has attempted to analyze Pakistani climate. Perhaps, the difference is that these other studies have considered only "shorter" time periods when studying Pakistani climate? And, yet some of the studies mentioned have analyzed multiple decades worth of data. Is what sets the present study apart the fact that it assesses changing climatic characteristics over a century? At first read, it seems that one important contribution of this work is that it tries to attribute the changes in aridity to precipitation and PET over different seasons. Perhaps this could be mentioned in the introduction.

It is also interesting that previous studies seem to suggest an increase in aridity, which

is not what this paper finds. It might be worth stating explicitly what time periods these studies considered or what methods they used so that the reader might gain some insight into why the present study finds such different conclusions. Comparisons between this and previous work could also be embellished in the discussion/conclusions.

\*\*\*

METHODS: I am a bit confused as to what a moving window of 50-years with 11-year interval is. Does this mean averages consist of 11 years of data? Why are only 50 years considered in the window if a century is available? (Is it because one hopes to analyze the transient nature of the trend, e.g. Section 4.5?) This could be mentioned first in the Methods for greater clarity.

Also, it would help the reader if the paper explained how the modified Mann-Kendall test better allows one to detect trends in the presence of natural variability. Is the natural variability assumed autocorrelated?

Generally, it might aid the reader if the Methods explicitly mentioned which tests were used for which experiments (i.e. To detect significant changes over the full time period, Sen's slope was used. To evaluate variability in the trend over the course of the century, an 11-year moving average was applied to 5-decade windows of data. . .)

Moreover, for a reader less familiar with the statistical tests used, a somewhat more detailed explanation of the variables and their physical significance could be useful. As one example, it is not entirely clear what "d" and the "critical value" are in Section 3.4.

Page 11 Line 3: I think Sen's "slopes" is meant.

\*\*\*

SECTION 4.3 seems somewhat misleading; either that or the legend for Figure 4 is not sufficiently detailed for the reader. The text suggests only a few locations are experiencing change, while the figure (Fig 4) shows large regions colored in ways that indicate change is occurring. Do the symbols (plusses and minuses?) in Fig 4 indicate

some type of significance? What is the difference between bold and light symbols (e.g. it is almost as if the shading was layered on top of the symbols accidentally in panel f?). The symbols should be described in the caption and perhaps made bigger (at least the minuses) for easier interpretation. Generally, throughout the manuscript, figure captions could include more details about the symbols and their significance, units, data source and/or years, etc.
* * *
SECTION 4.4: It is a bit confusing that Figure 5 uses five aridity "trend classes" that do not correspond to the climatological classes (e.g. arid, semi-arid, etc) introduced earlier in the text. Perhaps some clarifying language here would help. Also, it is not at all obvious that positive numbers indicate decreases in aridity. Either that, or the descriptions for either the annual trends or Kharif appear to be incorrect. The physical meaning of positive and negative numbers in Figure 5 should be made very clear for the reader. This could be explained initially in Section 3.1. (i.e. that higher AI indicates more humid not arid conditions) and re-mentioned again in Section 4.4.
* * *
SECTION 4.5 should be a bit cautious about attributing causation using words like "triggered." Figures 6 and 7 show a correspondence (correlation) between the areal change in precipitation and aridity but give no evidence of geographic overlap, which is presumably necessary for one factor to influence the other. In some ways, Figures 4 and 5 show the geographic correspondence much more clearly. That said, I do like the way Figures 6 and 7 show that trends changed in hand in hand, indicating, at least across Pakistan, that these climatic factors were influenced by the same large-scale transient controls. I would recommend that Figures 6 and 7 revisit their color choices. It is often most intuitive to use red for drying and blue for moistening. Changes in grid number clearly affect drying and moistening across Pakistan, but perhaps it is just simpler to use a more "neutral" color palette? Both the main text and the Figure

captions could better explain whether changes in positive trend grid numbers imply a more or less arid Pakistan.

\*\*\*

The DISCUSSION does a nice job of mentioning previous studies that attempted to attribute changes in aridity to factors such as precipitation or temperature. However, it says little about the fact that these previous studies tended to find that Pakistan's aridity is increasing. In contrast, the current manuscript finds that a much greater spatial area is wetting compared to drying. A bit greater emphasis on and discussion of this discrepancy seems warranted.

\*\*\*

CONCLUSIONS: I don't see where Kharif precipitation is decreasing over a large area in the east. As far as I can tell, there is a large increase in the northeast, adjacent to a very small area of decrease.

Also, I am surprised that the conclusion fails to mention the fact that this study finds that Pakistan is becoming less dry overall. "Changes in aridity" is simply not clear enough. I would encourage the paper to say, instead, "getting wetter/drier" or "less/more arid," etc.

The last paragraph is quite vague and could be more specific for a concluding paragraph.

\*\*\*

MINOR COMMENTS

The paper should be carefully reviewed for grammar and typos.

As far as I know, "On the other hand" is the phrase typically used (rather than "on the other side").

Page 23 Line 6: The second point should just say, "Most of the country…"
* * *

---

## Referee Comment (RC3) · Anonymous Referee #3 · 15 Mar 2019

Comments on "Spatiotemporal Changes in Aridity of Pakistan during 1901–2016" submitted by Kamal Ahmed et al. to Hydrology and Earth System Science

General comments In this manuscript, the authors investigated spatial distribution and temporal trend of precipitation, potential evapotranspiration, and aridity in Pakistan for the 20th and 21st centuries. They used GPCC and CRU PET datasets, allowing them to conduct long-term analyses. Several non-parametric statistics such as Mann-Kendall test and Sen's slope estimator were used. These tend analyses were conducted for annual and two cropping seasons (Kharif and Rabi); most readers are not familiar with this cropping scheme but seems unique in this study. They found that some significant changes in hydrological regime in this region occurred between 1971 and 1980. I know that Pakistan is one of the highly-populated regions and its hydro-

logical regime shift would have serious impacts, directly by altering available water and indirectly by affecting agricultural yields. In this regard, this kind of study is meaningful for planning sustainable society. Nevertheless, this study obtained, in my view, quite data-specific results. In general, long-term data such as GPCC and CRU are subject to uncertainties especially in the early period when observational data so sparse. Recent studies use multiple datasets to examine the consistency and difference, in order to obtain robust results. Indeed, no range of uncertainty or confidence interval was shown for the results obtained in this study. The use of non-parametric statistics looks reasonable. However, the hydrological regime shift obtained by these analyses were not adequately discussed. In Discussion (Page 22 Line 5), the authors related the regime shifts to the Asian monsoon, but no sufficient evidence was presented. I guess some human impacts, such as land-use conversion and overuse of ground water, could be responsible. Finally, I felt insufficient about the lack of discussion about the impacts of aridity change on human dimensions such as agriculture. Although this is not the main topic of the manuscript, I recommend adding some discussion about the impacts of increasing or decreasing aridity in this area. Finally, I can't recommend the manuscript as a candidate for publication in the present form. Please look my specific points.

Specific points Figure 4: What the dots in the panels represent? I guess it means significance of trend, but please clarify.

---

## Author Comment (AC1) · 29 Apr 2019

Comment The present article deals with aridity in Pakistan, certainly an important issue that merits publication in HESS. However, in its present form, this MS has too many drawbacks and a very limited benefit to the readers. Reply Thanks for your valuable time and constructive comments on our manuscript. The manuscript has now been revised according to the comments. The details of the revisions made are given under each comment. Revisions are marked in Red.

Comment This is due to several reasons such as, confusing definitions of the two seasons Rabi and Kharif (p.23, l.21-25). Once Rabi is defined as Nov.-May and later as Dec-Mar.; Kharif is defined as Apr.-Oct. and then as Jun.-Sep. To which definition

one should refer? Reply In order to avoid the confusion, we have defined both in section "2.1 Description of the Study Area" as below: "Rabi and Kharif are the two major cropping seasons of Pakistan (Chaudhry and Rasul, 2004). The Rabi season commences in November and finishes in May while the Kharif season starts in April and finishes in October (Nabeel and Athar, 2018). Besides cropping seasons, there are two major rainy seasons, i.e. winter and monsoon which coincide with the Rabi and Kharif season. Winter precipitation begins in December and lasts till March is important for Rabi crops while Monsoon precipitation begins in June and lasts till September is important for Kharif crops. Winter precipitation occurs due to the moist wind from the Mediterranean Sea in the west and north of Pakistan (Hussain and Lee, 2014). On the other hand, monsoon precipitation occurs due to the moist wind from the Bay of Bengal which contributes 60% of total precipitation of the country (Sheikh, 2001). The agro-economy and the livelihood of farmers constituting 43% of the total population of Pakistan depend on winter and monsoon precipitation (Ahmed et al., 2018a)."

Comment Non-comparable presented maps and partial information in some charts. Figure 1 - I assume that the six different height categories were selected in order to present equal areas in each category. This caused that range of each category is arbitrary and unusual. However, I could cope with this figure as we don't have to compare it to other maps. It is more severe with the rest of the maps. Figure 2 - The same problem as with the previous. It is impossible to compare among the precipitation maps (2a-2c) or the PET maps (2d-2f) in the different seasons. Furthermore, I suggest the authors to present the precipitation and the PET in the different seasons (2b-2c and 2e-2f, respectively) as a percentage of the annual totals and not as absolute values. This will be more explicit. In their present form, these maps are completely useless. Reply Thanks for your suggestion. We have revised figure 2b-2c and 2e-2f as a percentage of annual totals of precipitation and PET as below:

Figure 2. Spatial distribution of (a) annual, (b) Rabi and (c) Kharif precipitation; and (d) annual, (e) Rabi and (f) Kharif potential evapotranspiration in Pakistan

Comment Section 4.5 and Figure 6 and 7 – The authors present a moving average of 11 years of aridity, precipitation and PET trends. I don't understand why they chose increments of 50 years, why not calculate a moving average of 11 yrs. (or any other duration) for the entire period? Such a calculation would result in a less "fuzzy" behavior of the trends and enable to better locate the drier or wetter periods. Apparently there is no reason to assume that trends change with increments of 50 years. Reply To omit the confusion, we have added a new sub-section (3.4) to describe the issue of 50 years moving window with 11- years interval in method section for clarity as below: "3.4 Relationship of Aridity Trends with Precipitation and PET The relationships of precipitation and PET with aridity are assessed using a moving window of 50-year with 11-year interval over the study period, i.e., 1901-1950, 1912-1961, 1923-1972, 1934-1983, 1945-1994, 1956-2005 and 1967-2016. The main purpose of considering a 50-year window is to decipher the changing pattern in the relationship over the study period. The 11-year interval was considered to assess the relationship for the whole period (1901-2016)."

Besides 50-years moving window, we also assessed the aridity, precipitation and PET trends for the period 1901 to 2016 and presented results in section 4.2.

Comment Furthermore, a comparison of Figures 6 and 7 reveals that the trends during Kharif are by far more important in determining the entire year trends. Figure 7 is misleading as the vertical axes of a, b and c (Rabi) are different from those of d, e and f (Kharif) and give the impression of trends of the same order of magnitude in both period, which is not the case. Reply Thanks for the comment. To avoid the confusion, we have revised figure by keeping same vertical axis in figure 8 and 9 (previously figures 6 and 7). Furthermore reviewer 2 suggested using neutral color as below:

Figure 8. Number of grids where annual aridity, precipitation and PET are changed significantly during different time periods

Figure 9. Number of grids where annual Rabi and Kharif aridity, precipitation and PET

are changed significantly during different time periods

Comment The results presented in Figure 8 and Table 1 contradict the postulated in the introduction regarding increased aridity in Pakistan (p.2) and the results cited from Haider and Adman (2014). Overall there is no tendency over the majority of the territory (88%) and in those regions with a tendency it is mainly towards a reduction in aridity. Therefore, large parts of the introduction are irrelevant. Concluding that aridity depends mainly on precipitation (p.23, l-14-15) is very trivial. Reply Thanks for your comment. We have addressed the issue in introduction section as below: "Both increasing and decreasing trend in aridity has been reported in different regions of the world due to climate change. Several studies reported an increase in aridity in global (Dai, 2013;Trenberth et al., 2014) and regional (Ramarao et al., 2018;Jiao et al., 2016) scales. On the other hand, decrease in aridity is also reported in USA (Finkel et al., 2016), China (Yin et al., 2018) and some regions of Iran (Tabari and Talaee, 2013). In recent years, an increase in aridity in some regions of Pakistan has been reported (Haider and Adnan, 2014). However, it was just anticipation based on the assumption that rising temperature has intensified PET and thus an increase in aridity. The magnitude of temperature rises and the changes in regional precipitation pattern determines the changes in the aridity of an area. Therefore, it is required to assess the changes in aridity in regional scale considering the changes in both temperature and precipitation due to global warming.

Also to this, the issue of reduction in aridity is discussed in discussion section as below: "The aridity is found to increase (drier) and decrease (wetter) in different regions and seasons with the changes in precipitation and PET. Overall, 11.75%, 7.57%, and 9.66% areas are found to shift to wetter while 0.52%, 4.44%, and 0.52% areas to drier condition for annual, Rabi and Kharif respectively. It is important to mention that a large area has a wetter trend in recent years particularly in the semi-arid or sub-humid regions which mean more area become wetter in recent years. However, some areas in the arid region are found to become drier. This indicates that few dry regions are

becoming drier and a large relatively wet area is becoming wetter. A similar finding has been reported by Liu et al. (2018b) in neighbouring China. Overall, a large area in the northeast of Pakistan has become wetter and a few locations in the south become dried during 1901 - 2016." Also in conclusion: "(6) Overall, there is a wetting tendency over a large area in the northeast and drying tendency at few locations in the southwest. Therefore, it can be remarked that Pakistan has become wetter from 1901 to 2016."

Besides, we re-structured the sentences to mention in result that the changes in precipitation have a higher impact compared to PET in determining the aridity of Pakistan. In conclusion, we mentioned: "The time-varying trends in aridity reveal that the influence of precipitation is high on the aridity compared with PET." Comment The authors should consider editing the text by a native English speaker. Reply Thanks for your comment. The manuscript has now been proofread by a native English speaker. All the grammatical mistakes have been corrected. The English language has been improved.

Please also note the supplement to this comment:
https://www.hydrol-earth-syst-sci-discuss.net/hess-2018-642/hess-2018-642-AC1-supplement.pdf

―――――――――――――――――――

**Annual Precipitation (mm)**
38 - 158
159 - 253
254 - 385
386 - 585
586 - 876
877 - 1277
1278 - 2390

(a)

**Annual Potential Evapotranspiration (mm)**
664 - 986
987 - 1205
1206 - 1418
1419 - 1614
1615 - 1810
1811 - 2099
2100 - 2529

(d)

**Rabi Precipitation (%)**
< 20
21 - 40
41 - 60
61 - 80
> 80

(b)

**Rabi Potential Evapotranspiration (%)**
< 30
31 - 35
36 - 40
> 45

(e)

**Kharif Precipitation (%)**
< 20
21 - 40
41 - 60
61 - 80
> 80

(c)

**Kharif Potential Evapotranspiration (%)**
< 60
61 - 65
66 - 70
> 75

(f)

**Fig. 1.**

Interactive
comment

[Figure]

**Fig. 2.**

[Figure]

Fig. 3.

**Rabi** (left column):
- (a) Aridity Trends — Positive Trend, Negative Trend
- (b) Precipitation Trends — Positive Trend, Negative Trend
- (c) PET Trends — Positive Trend, Negative Trend

**Kharif** (right column):
- (d) Aridity Trends — Positive Trend, Negative Trend
- (e) Precipitation Trends — Positive Trend, Negative Trend
- (f) PET Trends — Positive Trend, Negative Trend

X-axis categories: 1901-1950, 1912-1961, 1923-1972, 1934-1983, 1945-1994, 1956-2005, 1967-2016
Y-axis: No of Grids

---

## Author Comment (AC2) · 29 Apr 2019

Comment This paper analyzes spatial patterns of aridity across Pakistan and attempts to attribute spatial and temporal changes in these patterns to changes in precipitation and potential evapotranspiration (PET). The paper shows that Pakistan is becoming less arid over a large region (the Northeast), most likely due to increases in precipitation, which seems to contrast with the findings of previous studies. Overall, I found the variety of statistical tests performed an interesting way to attribute the change in aridity to changes in precipitation with some degree of confidence. However, I was surprised that the paper failed to emphasize the clear geographic correspondence between changes in aridity and precipitation trends shown in Figures 4-5. My biggest criticism of the paper is the lack of clarity in describing the physical meaning of the

results. An increasing aridity index (AI) that indicates decreasing aridity is confusing enough. Add to that the desire to communicate changing trends in AI, and one can see how a reader becomes quickly confused. The paper would be greatly improved if it simply stated from time to time whether the results indicate that Pakistan is become more or less arid. There are also a number of places (noted below) where figure captions should contain more detail. My second biggest concern is that the paper fails to discuss its main finding, that Pakistan has become less arid, in the context of previous studies, which largely indicate that Pakistan has recently dried. The fact that this result seems to contrast with previous investigations ought to be discussed.

Reply Thank you very much for your valuable time and constructive comments on our manuscript. We have addressed your major concerns carefully in the revised manuscript. Additionally, we tried to address all your comments which are detailed in comments given below. We hope you will find the revised paper suitable for publication. Comment INTRODUCTION: The second paragraph of the Introduction lists many studies that have evaluated Pakistan's climate. A brief synopsis of their findings (not just their methodology) seems warranted. Reply Thanks for your comment; we have revised related text by adding the findings of studies as below: "Pakistan located in South Asia has a complex terrain with limited water resources. Several attempts have been made to classify the aridity and climate of Pakistan based on different climate variables and methods (Bharuqha and Shanbhag, 1956;Oliver et al., 1978;Shamshad, 1988;Chaudhry and Rasul, 2004;Hussain and Lee, 2009;Zahid and Rasul, 2011;Sarfaraz, 2014;Sarfaraz et al., 2014;Haider and Adnan, 2014). Bharuqha and Shanbhag (1956) classified the climate of a station (Hyderabad) based on the fraction of precipitation to evaporation for the period 1926−1940 and found that Hyderabad has an arid (desert) climate. Oliver et al. (1978) applied clustering approach for climate classification using meteorological data from 53 stations. The results of the study showed that Pakistan has nine climate regimes where most of the area falls under arid climate. Chaudhry and Rasul (2004) used Thornthwaite's precipitation effectiveness index (PEI) for the estimation of annual and seasonal aridity for the period 1961-1990 using temperature data of 50 stations. The results showed that around 75% of the land has arid climate while only a small area in the north-eastern plain has a sub-humid climate. Hussain and Lee (2009) classified the climate using factor and cluster analysis utilising 26 years (1980-2006) rainfall and temperature records of 32 stations. The study concluded that the land of Pakistan could be divided into six regions based on the topology of the country. Haider and Adnan (2014) used several aridity indices (De Martonne Aridity index, Erinc Aridity index, Thornthwaite's PEI, UNESCO Aridity index and Thornthwaite Moisture index) to classify the climate of Pakistan based on records of 54 stations for the period 1961-2009. Their study reported that around 75 to 85% of the land of the country belongs to the arid climate and less than 10% of land in the north belongs to the humid climate. Sarfaraz (2014) used principal component analysis for the sub-regional classification of Pakistan's winter precipitation using 35 station data from 1976 to 2005 and reported six sub-regions of winter precipitation in Pakistan. Sarfaraz et al. (2014) used Köppen classification to classify the climate based on 59 stations data for the period 1981 to 2010 and showed that 75% of the country has arid to semi-arid climate. Recently, Nabeel and Athar (2018) classified the climate based on wet and dry spell using 46 stations data for the period 1976 - 2007. They reported that 66% of the country belongs to the arid climate while only 4% belongs to the humid climate."

Comment Furthermore, it seems a bit contradictory to end the second paragraph by saying that "no attempt has been made. . .to assess the changing characteristics of arid climatology. . ." and then begin the third by saying "In recent years, an increase in aridity is reported. . ." Clearly, someone has attempted to analyze Pakistani climate. Perhaps, the difference is that these other studies have considered only "shorter" time periods when studying Pakistani climate? And, yet some of the studies mentioned have analyzed multiple decades worth of data. Is what sets the present study apart the fact that it assesses changing climatic characteristics over a century? At first read, it seems that one important contribution of this work is that it tries to attribute the changes in aridity to precipitation and PET over different seasons. Perhaps this could be mentioned in the introduction. Reply Thanks for the suggestion; we have revised related text as below: "Even though several studies have been conducted for the classification of climate using aridity indices, there is still no comprehensive study to assess the long-term trends in the aridity of Pakistan in different seasons (annual, Kharif and Rabi). Furthermore, no study has been conducted to determine the impacts of climate change on aridity, particularly the influence of different climate variables like precipitation, temperature and potential evapotranspiration on aridity in different seasons. Both increasing and decreasing trend in aridity has been reported in different regions of the world due to climate change. Several studies reported an increase in aridity in global (Dai, 2013;Trenberth et al., 2014) and regional (Ramarao et al., 2018;Jiao et al., 2016) scales. On the other hand, decrease in aridity is also reported in USA (Finkel et al., 2016), China (Yin et al., 2018) and some regions of Iran (Tabari and Talaee, 2013). In recent years, an increase in aridity in some regions of Pakistan has been reported (Haider and Adnan, 2014). However, it was just anticipation based on the assumption that rising temperature has intensified PET and thus an increase in aridity. The magnitude of temperature rises and the changes in regional precipitation pattern determines the changes in the aridity of an area. Therefore, it is required to assess the changes in aridity in regional scale considering the changes in both temperature and precipitation due to global warming."

Comment It is also interesting that previous studies seem to suggest an increase in aridity, which is not what this paper finds. It might be worth stating explicitly what time periods these studies considered or what methods they used so that the reader might gain some insight into why the present study finds such different conclusions. Comparisons between this and previous work could also be embellished in the discussion/conclusions. Reply Thanks for your comment. All previous studies classified the climate of Pakistan based on different aridity indices; none of them has assessed the trends in aridity or attributed aridity changes with climate variables. This is the first study where we classified aridity in different seasons, assessed aridity and climate variable trends over a longer temporal scale, attributed aridity changes with precipitation

and PET. In one of the previous study Haider and Adnan, 2014 reported that aridity has increased over some regions Pakistan based on the speculation that increase in temperature and decrease in precipitation are causing aridity changes. However, their study was only limited to the classification of aridity using different aridity indices. We have discussed this in introduction section as below: "In recent years, an increase in aridity in some regions of Pakistan has been reported (Haider and Adnan, 2014). However, it was just anticipation based on the assumption that rising temperature has intensified PET and thus an increase in aridity. The magnitude of temperature rises and the changes in regional precipitation pattern determines the changes in the aridity of an area. Therefore, it is required to assess the changes in aridity in regional scale considering the changes in both temperature and precipitation due to global warming."

To address the issue of wetting tendency over Pakistan, we have restructured the following paragraph to make it clearer: In discussion section: "The aridity is found to increase (drier) and decrease (wetter) in different regions and seasons with the changes in precipitation and PET. Overall, 11.75%, 7.57%, and 9.66% areas are found to shift to wetter while 0.52%, 4.44%, and 0.52% areas to drier condition for annual, Rabi and Kharif respectively. It is important to mention that a large area has a wetter trend in recent years particularly in the semi-arid or sub-humid regions which mean more area become wetter in recent years. However, some areas in the arid region are found to become drier. This indicates that few dry regions are becoming drier and a large relatively wet area is becoming wetter. A similar finding has been reported by Liu et al. (2018b) in neighbouring China. Overall, a large area in the northeast of Pakistan has become wetter and a few locations in the south become dried during 1901 - 2016." Also in conclusion: "(6) Overall, there is a wetting tendency over a large area in the northeast and drying tendency at few locations in the southwest. Therefore, it can be remarked that Pakistan has become wetter from 1901 to 2016."

Comment METHODS: I am a bit confused as to what a moving window of 50-years with 11-year interval is. Does this mean averages consist of 11 years of data? Why

are only 50 years considered in the window if a century is available? (Is it because one hopes to analyze the transient nature of the trend, e.g. Section 4.5?) This could be mentioned first in the Methods for greater clarity. Reply Thanks for your comment. In 50-years moving window with 11-year interval, we have assessed aridity using 50 years aridity over different time periods (1901-1950, 1912-1961, 1923-1972, 1934-1983, 1945-1994, 1956-2005, 1967-2016) starting from 1901-1950 with interval of 11 years. 11 years was considered to cover fifty years from 1901 to 2016. Following your suggestion, we have describes the issue in method section for clarity as below: "3.4 Relationship of Aridity Trends with Precipitation and PET The relationships of precipitation and PET with aridity are assessed using a moving window of 50-year with 11-year interval over the study period, i.e., 1901-1950, 1912-1961, 1923-1972, 1934-1983, 1945-1994, 1956-2005 and 1967-2016. The main purpose of considering a 50-year window is to decipher the changing pattern in the relationship over the study period. The 11-year interval was considered to assess the relationship for the whole period (1901-2016)."

Comment Also, it would help the reader if the paper explained how the modified Mann-Kendall test better allows one to detect trends in the presence of natural variability. Is the natural variability assumed autocorrelated? Reply Thanks for your comment. We have describes modified Mann Kendall trend as below: "3.3 Modified Mann-Kendall (MMK) Test In the MMK test (Hamed, 2008), the significance in the trend is first computed by applying classical MK test. The MK test statistics (S) for time series with n data points can be calculated as: (2) Where and are sequential data and is calculated as below: (3) The standardised test static () is then calculated from the variance of S as, (5) The null hypothesis on no trend is rejected at a confidence interval of 95% if .The MMK test is conducted when the null hypothesis of no trend is rejected. For this purpose, the existing trend in time series data is removed, and the data are ranked. The equivalent normal variants of ranked data (Ri) are calculated as, (6) Where is the inverse standard normal distribution function. The Hurst coefficient (H) is estimated by maximising the log-likelihood function. If H is found significant, the biased variance

of S is calculated as, (7) Where is the auto-correlation function for given H. The unbiased estimate is calculated as, (8) Where B is a function of H as below: (9) Where the coefficients, , , , and are the functions of the sample size n, which can be found in Hamed (2008). The significance of the MMK test is computed by using in place of in equation (5). Natural variability appears as long-term autocorrelation in data. This has been mentioned by restructuring the sentence in Introduction as below: "It is expected that the use of MMK test would provide the changes in aridity due to global warming by eliminating the effect of natural variability of climate which infested as a long-term autocorrelation in time series."

Comment Generally, it might aid the reader if the Methods explicitly mentioned which tests were used for which experiments (i.e. To detect significant changes over the full time period, Sen's slope was used. To evaluate variability in the trend over the course of the century, an 11-year moving average was applied to 5-decade windows of data. . .) Reply Thanks for your suggestion, we have added following text in the method section to improve the readability: "The procedure used for the assessment of the changes in the characteristics of aridity in Pakistan is outlined below: 1)The aridity is estimated as the ratio of precipitation to PET at each GPCC/CRU grid point for all the years during 1901 – 2016. The aridity values are estimated separately for annual, Rabi and Kharif seasons. 2)Sen's slope estimator is used to estimate the rate of change in precipitation, PET and aridity in annual, Rabi and Kharif seasons for the period 1901 – 2016. 3)The MMK trend test is used to evaluate the significance of the change in precipitation, PET and aridity for all the seasons. 4)The influence of precipitation and PET on aridity is assessed for different 50-year with an interval of 11-year over the period 1901 – 2016. 5)The shift in the aridity from one aridity class to another between two periods, 1901 – 1950 and 1967 – 2016 is mapped to assess the changes in areal extent of different arid classes. 6)The Pettitt's test is used to detect the change points in aridity, precipitation and PET in Pakistan."

Comment Moreover, for a reader less familiar with the statistical tests used, a some-

what more detailed explanation of the variables and their physical significance could be useful. As one example, it is not entirely clear what "d" and the "critical value" are in Section 3.4. Reply Thanks for your suggestion; we have described all methods in more details in the revised manuscript.

Comment Page 11 Line 3: I think Sen's "slopes" is meant. Reply Sorry for the mistake. Correction is made.

Comment SECTION 4.3 seems somewhat misleading; either that or the legend for Figure 4 is not sufficiently detailed for the reader. The text suggests only a few locations are experiencing change, while the figure (Fig 4) shows large regions colored in ways that indicate change is occurring. Do the symbols (plusses and minuses?) in Fig 4 indicate some type of significance? What is the difference between bold and light symbols (e.g. it is almost as if the shading was layered on top of the symbols accidentally in panel f?). The symbols should be described in the caption and perhaps made bigger (at least the minuses) for easier interpretation. Generally, throughout the manuscript, figure captions could include more details about the symbols and their significance, units, data source and/or years, etc. Reply Thanks for your suggestion; we have prepared larger figures with legends to improve the readability. We have increased the size of plus and minuses (to indicate the significance) on the figures. Additionally we have revised the captions of figures for easier interpretation. Furthermore, we have also revised the related text to make it more clear for the reader as below: "The sen's slope is used to assess the magnitude of change in precipitation and PET for all the seasons at all the 350 grid points over Pakistan to prepare the corresponding maps as shown in Figures 4 to 6. The significance increasing/decreasing trends estimated using MMK test at 95% level of confidence are presented using the plus (+) and minus (-) signs in the figures. The increase in precipitation indicates a wetter and the decrease a drier condition, while an increase in PET indicates a drier and decrease a wetter condition. Figure 4a shows that annual precipitation is increasing significantly over a large area in the northeast and at a few places in the far north, while it is decreasing significantly at

a few places in the south and three locations near the foothills of Himalaya. It is worth to mention that precipitation is decreasing at a few locations near the foothills of the Himalaya where precipitation is highest in Pakistan (Figure 2a). The spatial distributions of the trends in annual PET are shown in Figure 4b. The annual PET in Pakistan is increasing (high evaporation rates) in the southeast corner and decreasing (low evaporation rates) at a few grid points scattered in the center and north-western parts where precipitation is usually high, and the temperature is low.

Figure 4. Spatial distribution of the trends in annual (a) precipitation and (b) PET in Pakistan estimated using modified Mann-Kendall (MMK) test. The plus (+) and minus (-) sign indicates increasing and decreasing trend at 95% confidence level respectively.

Figure 5a shows the spatial patterns in the trend of Rabi precipitation. The precipitation during Rabi is found to increase significantly at a few grid points in the north and two grid points in the east while decreasing significantly at two locations in the south. It can be observed that there is a non-significant decreasing tendency in Rabi precipitation over a large in the south. The PET in Rabi (Figure 5b) is found to increase significantly (high evaporation rates) over a large area in the southeast and the southwest, while it is not found to decrease significantly at any location.

Figure 5. Spatial distribution of the trends in Rabi (a) precipitation and (b) PET in Pakistan estimated using modified Mann-Kendall (MMK) test. The plus (+) and minus (-) sign indicates increasing and decreasing trend at 95% confidence level respectively.

The Kharif precipitation (Figure 6a) is found to increase significantly in the northeast and at two grid points in the north. The significant decreasing trend in Kharif precipitation is also observed over a large area in the southwest and at a few grid points near the foothills of Himalaya. Overall, the spatial patterns in annual and Kharif precipitation trends are found very similar. The spatial distribution of PET trends in Kharif is displayed in Figure 6b. The figure shows a significant decrease in PET over a large area in the northeast and decreases at two grid points in the south.

Figure 6. Spatial distribution of the trends in Kharif (a) precipitation and (b) PET in Pakistan estimated using modified Mann-Kendall (MMK) test. The plus (+) and minus (-) sign indicates increasing and decreasing trend at 95% confidence level respectively.

Comment SECTION 4.4: It is a bit confusing that Figure 5 uses five aridity "trend classes" that do not correspond to the climatological classes (e.g. arid, semi-arid, etc) introduced earlier in the text. Perhaps some clarifying language here would help. Also, it is not at all obvious that positive numbers indicate decreases in aridity. Either that, or the descriptions for either the annual trends or Kharif appear to be incorrect. The physical meaning of positive and negative numbers in Figure 5 should be made very clear for the reader. This could be explained initially in Section 3.1. (i.e. that higher AI indicates more humid not arid conditions) and re-mentioned again in Section 4.4. Reply Thank you very much for your comment and suggestion. We understand that there was a confusion in interpretation of aridity values on the figure and text. In order to avoid the confusion in the interpretation of aridity trends, we have used word "drier and wetter conditions" which was suggested by the reviewer in earlier comments. The same is also mentioned in method section. The issue of aridity classes is also addressed in the revised text 4.4 as below: "The Sen's Slope method was used to estimate the changes in aridity values calculated using UNESCO method and the MKK test was used to determine the significance of the change at 95% level of confidence. The changes in aridity index are found in the range of -0.0039 to 0.0060 for Pakistan (Figure 7). The values were divided into five classes using natural break algorithm available in ArcGIS 10.3. The plus sign in the figure indicates a significant reduction in aridity (wetter condition) while the minus sign indicates a significant increase in aridity (drier condition). Figure 7a shows that the mean annual aridity has a significant wetter trend over a large area in the northeast and a significant drier trend at a few locations in the south. The aridity trends in Rabi (Figure 7b) shows a significant drier trend in the southwest, at two grid points in the center and in the south. Significant wetter conditions during Rabi are also observed at a few grid points in the north. The aridity trend in Kharif (Figure 7c) is found to follow similar patterns of annual aridity trend.
Significant wetter tend is noticed over a major area in the northeast and at a few grid points in the north while drier trend at few locations in the southwest and southern corner of the country. Overall, the results reveal a wetter trend over a major portion in the northeast and drier trend at few locations in the southwest.

Figure 7. Spatial distribution of the trends in (a) annual (b) Rabi and (c) Kharif aridity over Pakistan estimated using modified Mann-Kendall (MMK) test. The plus (+) and minus (-) sign indicates increasing and decreasing trend at 95% confidence level respectively.

Comment SECTION 4.5 should be a bit cautious about attributing causation using words like "triggered." Figures 6 and 7 show a correspondence (correlation) between the areal change in precipitation and aridity but give no evidence of geographic overlap, which is presumably necessary for one factor to influence the other. In some ways, Figures 4 and 5 show the geographic correspondence much more clearly. That said, I do like the way Figures 6 and 7 show that trends changed in hand in hand, indicating, at least across Pakistan, that these climatic factors were influenced by the same large-scale transient controls. I would recommend that Figures 6 and 7 revisit their color choices. It is often most intuitive to use red for drying and blue for moistening. Changes in grid number clearly affect drying and moistening across Pakistan, but perhaps it is just simpler to use a more "neutral" color palette? Both the main text and the Figure captions could better explain whether changes in positive trend grid numbers imply a more or less arid Pakistan.

Reply Thanks for the comment and suggestion. We have replaced word "triggered" in the manuscript. We have used "neutral" color palette for Figure 8 and 9 (previously figures 6 and 7) as below:

Figure 8. Number of grids where annual aridity, precipitation and PET are changed significantly during different time periods

Figure 9. Number of grids where annual Rabi and Kharif aridity, precipitation and PET

are changed significantly during different time periods

Comment The DISCUSSION does a nice job of mentioning previous studies that attempted to attribute changes in aridity to factors such as precipitation or temperature. However, it says little about the fact that these previous studies tended to find that Pakistan's aridity is increasing. In contrast, the current manuscript finds that a much greater spatial area is wetting compared to drying. A bit greater emphasis on and discussion of this discrepancy seems warranted. Reply Thanks for your comment. All previous studies classified the climate of Pakistan based on different aridity indices; none of them has assessed the trends in aridity or attributed aridity changes with climate variables. This is the first study where we classified aridity in different seasons, assessed aridity and climate variable trends over a longer temporal scale, attributed aridity changes with precipitation and PET. In one of the previous study Haider and Adnan, 2014 reported that aridity has increased over some regions Pakistan based on the speculation that increase in temperature and decrease in precipitation are causing aridity changes. However, their study was only limited to the classification of aridity using different aridity indices. We have discussed this in introduction section as below: "In recent years, an increase in aridity in some regions of Pakistan has been reported (Haider and Adnan, 2014). However, it was just anticipation based on the assumption that rising temperature has intensified PET and thus an increase in aridity. The magnitude of temperature rises and the changes in regional precipitation pattern determines the changes in the aridity of an area. Therefore, it is required to assess the changes in aridity in regional scale considering the changes in both temperature and precipitation due to global warming."

To address the issue of wetting tendency over Pakistan, we have restructured the following paragraph to make it clearer: In discussion section: "The aridity is found to increase (drier) and decrease (wetter) in different regions and seasons with the changes in precipitation and PET. Overall, 11.75%, 7.57%, and 9.66% areas are found to shift to wetter while 0.52%, 4.44%, and 0.52% areas to drier condition for annual, Rabi and

[Figure]

Kharif respectively. It is important to mention that a large area has a wetter trend in recent years particularly in the semi-arid or sub-humid regions which mean more area become wetter in recent years. However, some areas in the arid region are found to become drier. This indicates that few dry regions are becoming drier and a large relatively wet area is becoming wetter. A similar finding has been reported by Liu et al. (2018b) in neighbouring China. Overall, a large area in the northeast of Pakistan has become wetter and a few locations in the south become dried during 1901 - 2016." Also in conclusion: "(6) Overall, there is a wetting tendency over a large area in the northeast and drying tendency at few locations in the southwest. Therefore, it can be remarked that Pakistan has become wetter from 1901 to 2016."

Comment CONCLUSIONS: I don't see where Kharif precipitation is decreasing over a large area in the east. As far as I can tell, there is a large increase in the northeast, adjacent to a very small area of decrease. Also, I am surprised that the conclusion fails to mention the fact that this study finds that Pakistan is becoming less dry overall. "Changes in aridity" is simply not clear enough. I would encourage the paper to say, instead, "getting wetter/drier" or "less/more arid," etc. The last paragraph is quite vague and could be more specific for a concluding paragraph. Reply We agree that there is a mistake in Kharif precipitation interpretation; we have restructured the related sentence as below: "The annual and Kharif precipitation of Pakistan is increasing in the northeast, while Rabi precipitation is increasing at a few grid points in the north." Following your comment we have mentioned that Pakistan has wetter trend as below in conclusion: "(6) Overall, there is a wetting tendency over a large area in the northeast and drying tendency at few locations in the southwest. Therefore, it can be remarked that Pakistan has become wetter from 1901 to 2016." We have revised whole paper and used the word "wetter and drier" for easier interpretation as suggested. The last paragraph is provided for the limitations of the study; we have revised it and shifted mentioned para at the end of discussion section.

MINOR COMMENTS Comment The paper should be carefully reviewed for grammar

and typos. As far as I know, "On the other hand" is the phrase typically used (rather than "on the other side") Reply Thanks for your comment. The manuscript has now been proofread by a native English speaker. The phrase "on the other side" is also replaced with "On the other hand" in the whole manuscript.

Comment Page 23 Line 6: The second point should just say, "Most of the country…" Reply Thanks, correction is made.

Please also note the supplement to this comment:
https://www.hydrol-earth-syst-sci-discuss.net/hess-2018-642/hess-2018-642-AC2-supplement.pdf

[Figure]

[Figure]

**Fig. 1.**

[Figure]

[Figure]

**Fig. 2.**

[Figure]

**Fig. 3.**

[Figure]

**Fig. 4.**

**Fig. 5.**

[Figure]

Fig. 6.

- (a) Aridity Trends — Positive Trend, Negative Trend
- (b) Precipitation Trends — Positive Trend, Negative Trend
- (c) PET Trends — Positive Trend, Negative Trend

Right column labeled **Kharif**:
- (d) Aridity Trends — Positive Trend, Negative Trend
- (e) Precipitation Trends — Positive Trend, Negative Trend
- (f) PET Trends — Positive Trend, Negative Trend

---

## Author Comment (AC3) · 29 Apr 2019

General Comments In this manuscript, the authors investigated spatial distribution and temporal trend of precipitation, potential evapotranspiration, and aridity in Pakistan for the 20th and 21st centuries. They used GPCC and CRU PET datasets, allowing them to conduct long-term analyses. Several non-parametric statistics such as Mann-Kendall test and Sen's slope estimator were used. These tend analyses were conducted for annual and two cropping seasons (Kharif and Rabi); most readers are not familiar with this cropping scheme but seems unique in this study. They found that some significant changes in hydrological regime in this region occurred between 1971 and 1980. I know that Pakistan is one of the highly-populated regions and its hydrological regime shift would have serious impacts, directly by altering available water and

indirectly by affecting agricultural yields. In this regard, this kind of study is meaningful for planning sustainable society. Reply Thank you for your time and valuable comments on our manuscript. Your suggestions helped us to improve the quality of the paper. We have carefully addressed all your comments in the revision of the paper. Revised text is highlighted in red.

Comment Nevertheless, this study obtained, in my view, quite data-specific results. In general, long-term data such as GPCC and CRU are subject to uncertainties especially in the early period when observational data so sparse. Recent studies use multiple datasets to examine the consistency and difference, in order to obtain robust results. Indeed, no range of uncertainty or confidence interval was shown for the results obtained in this study. Reply Thank you very much for your comment. The PET and Temperature data from 1901 are available only for CRU. Therefore, it was not possible to assess the uncertainty due to gridded data. Such analysis can be conducted in future based on the availability of data. We mentioned this as limitation of the study in end of discussion section and recommended as future work as below: "The gridded data used in this may cause uncertainty in the estimation of aridity and its trends. Other gridded data can be used in future to assess the uncertainty in the estimated trends in aridity. Besides, different aridity assessment methods can be used to compare the results."

Comment The use of non-parametric statistics looks reasonable. However, the hydrological regime shift obtained by these analyses was not adequately discussed. In Discussion (Page 22 Line 5), the authors related the regime shifts to the Asian monsoon, but no sufficient evidence was presented. I guess some human impacts, such as land-use conversion and overuse of ground water, could be responsible.

Reply The precipitation changes in the region is discussed in details in the revised manuscript. Following texts have been added for this purpose: "The changes in temporal patterns of aridity reveal that the major shift in aridity and rainfall occurred between 1971 and 1980. Global atmospheric moisture amount is found to increase after

1973 (Ross and Elliott, 2001). An increase in precipitation in many regions of the world is observed due to the increase in global moisture content (Trenberth, 1998). The present study suggests that precipitation of Pakistan has also changed during 1971-1980 which may be due to the increase in global atmospheric moisture after 1973. This has caused a shift in precipitation and aridity in Pakistan. Machiwal et al. (2017) reported a significant change in dry season precipitation in the period 1973–1975 in the hot arid region of India. Some'e et al. (2012) assessed the change points in precipitation in the eastern part of Iran bordering Pakistan and reported a shift in annual precipitation at some stations during 1981-1982. The results collaborate with the finding of precipitation and aridity shift in Pakistan. Many factors influence regional and local changes in precipitation including shift in monsoon circulation due to global climate change (IPCC, 2014), land use changes like the changes in forest cover and irrigated agriculture (Pielke, 2001) and aerosols in the atmosphere due to human activities (Guo et al., 2016). Studies related to anthropogenic activities on precipitation changes in Pakistan and nearby countries are very limited (Basistha et al., 2009). Previous studies suggested that global warming as the cause of the shift in precipitation pattern in the region (Duan et al., 2002;Gautam et al., 2009). The nature of the shift in rainfall regime over a large region which coincides with the increase in global atmospheric moisture suggests that global climate change may be the cause of the shift in precipitation and aridity of Pakistan. The present study suggests that the relative influence of precipitation and temperature on aridity determines its trends in the context of climate change. Aridity may decrease due to a small increase in precipitation in the regions where the influence of precipitation is higher on aridity. The gridded data used in this may cause uncertainty in the estimation of aridity and its trends. Other gridded data can be used in future to assess the uncertainty in the estimated trends in aridity. Besides, different aridity assessment methods can be used to compare the results."

Comment Finally, I felt insufficient about the lack of discussion about the impacts of aridity change on human dimensions such as agriculture. Although this is not the main topic of the manuscript, I recommend adding some discussion about the impacts of

increasing or decreasing aridity in this area.

Reply Thanks for your comment, we have added following paragraph in discussion section: "Pakistan is mainly an agriculture-based country where a notable portion of the population is associated with the agro-based economy. Haider and Adnan (2014) reported that changes in aridity could have a severe impact on the agricultural sector of Pakistan. They showed that some regions in the northeast of the country are becoming less arid while some of the regions in the south are becoming drier. It is pertinent to mention that southern regions of the country are highly prone to droughts (Ahmed et al., 2018b). Increase in drier conditions can have a severe impact on the agricultural-based economy of the south. Similarly, the agriculture of north-eastern regions can be benefitted by the wetter condition."

Comment Finally, I can't recommend the manuscript as a candidate for publication in the present form. Please look my specific points. Specific points Figure 4: What the dots in the panels represent? I guess it means significance of trend, but please clarify. Reply Thanks for your suggestion; we have prepared larger figures with legends to improve the readability. We have increased the size of plus and minuses (to indicate the significance) on the figures. Additionally we have revised the captions of figures for easier interpretation. Furthermore, we revised the related text to make it more clear for the readers as below: "The sen's slope is used to assess the magnitude of change in precipitation and PET for all the seasons at all the 350 grid points over Pakistan to prepare the corresponding maps as shown in Figures 4 to 6. The significance increasing/decreasing trends estimated using MMK test at 95% level of confidence are presented using the plus (+) and minus (-) signs in the figures. The increase in precipitation indicates a wetter and the decrease a drier condition, while an increase in PET indicates a drier and decrease a wetter condition. Figure 4a shows that annual precipitation is increasing significantly over a large area in the northeast and at a few places in the far north, while it is decreasing significantly at a few places in the south and three locations near the foothills of Himalaya. It is worth to mention that

[Figure]

precipitation is decreasing at a few locations near the foothills of the Himalaya where precipitation is highest in Pakistan (Figure 2a). The spatial distributions of the trends in annual PET are shown in Figure 4b. The annual PET in Pakistan is increasing (high evaporation rates) in the southeast corner and decreasing (low evaporation rates) at a few grid points scattered in the center and north-western parts where precipitation is usually high, and the temperature is low.

Figure 4. Spatial distribution of the trends in annual (a) precipitation and (b) PET in Pakistan estimated using modified Mann-Kendall (MMK) test. The plus (+) and minus (-) sign indicates increasing and decreasing trend at 95% confidence level respectively.

Figure 5a shows the spatial patterns in the trend of Rabi precipitation. The precipitation during Rabi is found to increase significantly at a few grid points in the north and two grid points in the east while decreasing significantly at two locations in the south. It can be observed that there is a non-significant decreasing tendency in Rabi precipitation over a large in the south. The PET in Rabi (Figure 5b) is found to increase significantly (high evaporation rates) over a large area in the southeast and the southwest, while it is not found to decrease significantly at any location.

Figure 5. Spatial distribution of the trends in Rabi (a) precipitation and (b) PET in Pakistan estimated using modified Mann-Kendall (MMK) test. The plus (+) and minus (-) sign indicates increasing and decreasing trend at 95% confidence level respectively.

The Kharif precipitation (Figure 6a) is found to increase significantly in the northeast and at two grid points in the north. The significant decreasing trend in Kharif precipitation is also observed over a large area in the southwest and at a few grid points near the foothills of Himalaya. Overall, the spatial patterns in annual and Kharif precipitation trends are found very similar. The spatial distribution of PET trends in Kharif is displayed in Figure 6b. The figure shows a significant decrease in PET over a large area in the northeast and decreases at two grid points in the south.

Figure 6. Spatial distribution of the trends in Kharif (a) precipitation and (b) PET in

Pakistan estimated using modified Mann-Kendall (MMK) test. The plus (+) and minus (-) sign indicates increasing and decreasing trend at 95% confidence level respectively.

Please also note the supplement to this comment:
https://www.hydrol-earth-syst-sci-discuss.net/hess-2018-642/hess-2018-642-AC3-supplement.pdf

―――――――――――――――――

[Figure]

**Fig. 1.**

[Figure]

**Fig. 2.**

[Figure]

**Fig. 3.**